# Globally Optimal Training of Generalized Polynomial Neural Networks with Nonlinear Spectral Methods

**A. Gautier, Q. Nguyen and M. Hein**
Department of Mathematics and Computer Science
Saarland Informatics Campus, Saarland University, Germany

## Abstract

The optimization problem behind neural networks is highly non-convex. Training with stochastic gradient descent and variants requires careful parameter tuning and provides no guarantee to achieve the global optimum. In contrast we show under quite weak assumptions on the data that a particular class of feedforward neural networks can be trained globally optimal with a linear convergence rate with our nonlinear spectral method. Up to our knowledge this is the first practically feasible method which achieves such a guarantee. While the method can in principle be applied to deep networks, we restrict ourselves for simplicity in this paper to one and two hidden layer networks. Our experiments confirm that these models are rich enough to achieve good performance on a series of real-world datasets.

## 1    Introduction

Deep learning [13, 16] is currently the state of the art machine learning technique in many application areas such as computer vision or natural language processing. While the theoretical foundations of neural networks have been explored in depth see e.g. [1], the understanding of the success of training deep neural networks is a currently very active research area [5, 6, 9]. On the other hand the parameter search for stochastic gradient descent and variants such as Adagrad and Adam can be quite tedious and there is no guarantee that one converges to the global optimum. In particular, the problem is even for a single hidden layer in general NP hard, see [17] and references therein. This implies that to achieve global optimality efficiently one has to impose certain conditions on the problem.

A recent line of research has directly tackled the optimization problem of neural networks and provided either certain guarantees [2, 15] in terms of the global optimum or proved directly convergence to the global optimum [8, 11]. The latter two papers are up to our knowledge the first results which provide a globally optimal algorithm for training neural networks. While providing a lot of interesting insights on the relationship of structured matrix factorization and training of neural networks, Haeffele and Vidal admit themselves in their paper [8] that their results are "challenging to apply in practice". In the work of Janzamin et al. [11] they use a tensor approach and propose a globally optimal algorithm for a feedforward neural network with one hidden layer and squared loss. However, their approach requires the computation of the score function tensor which uses the density of the data-generating measure. However, the data generating measure is unknown and also difficult to estimate for high-dimensional feature spaces. Moreover, one has to check certain non-degeneracy conditions of the tensor decomposition to get the global optimality guarantee.

In contrast our nonlinear spectral method just requires that the data is nonnegative which is true for all sorts of count data such as images, word frequencies etc. The condition which guarantees global optimality just depends on the parameters of the architecture of the network and boils down to the computation of the spectral radius of a small nonnegative matrix. The condition can be checked without running the algorithm. Moreover, the nonlinear spectral method has a linear convergence rate and thus the globally optimal training of the network is very fast. The two main changes compared to the standard setting are that we require nonnegativity on the weights of the network and we have to minimize a modified objective function which is the sum of loss and the negative total sum of the outputs. While this model is non-standard, we show in some first experimental results that the resulting classifier is still expressive enough to create complex decision boundaries. As well, we achieve competitive performance on some UCI datasets. As the nonlinear spectral method requires some non-standard techniques, we use the main part of the paper to develop the key steps necessary for the proof. However, some proofs of the intermediate results are moved to the supplementary material.

## 2  Main result

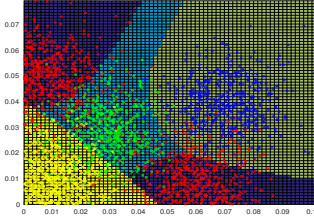

**Figure 1:** Classification decision boundaries in $\mathbb{R}^2$. (Best viewed in colors.)

In this section we present the algorithm together with the main theorem providing the convergence guarantee. We limit the presentation to one hidden layer networks to improve the readability of the paper. Our approach can be generalized to feedforward networks of *arbitrary* depth. In particular, we present in Section 4.1 results for two hidden layers.

We consider in this paper multi-class classification where $d$ is the dimension of the feature space and $K$ is the number of classes. We use the negative cross-entropy loss defined for label $y \in [K] := \{1, \ldots, K\}$ and classifier $f : \mathbb{R}^d \to \mathbb{R}^K$ as

$$L\big(y, f(x)\big) = -\log\left(\frac{e^{f_y(x)}}{\sum_{j=1}^{K} e^{f_j(x)}}\right) = -f_y(x) + \log\Big(\sum_{j=1}^{K} e^{f_j(x)}\Big).$$

The function class we are using is a feedforward neural network with one hidden layer with $n_1$ hidden units. As activation functions we use real powers of the form of a generalized polyomial, that is for $\alpha \in \mathbb{R}^{n_1}$ with $\alpha_i \geq 1$, $i \in [K]$, we define:

$$f_r(x) = f_r(w, u)(x) = \sum_{l=1}^{n_1} w_{rl} \Big( \sum_{m=1}^{d} u_{lm} x_m \Big)^{\alpha_l}, \tag{1}$$

where $\mathbb{R}_+ = \{x \in \mathbb{R} \mid x \geq 0\}$ and $w \in \mathbb{R}_+^{K \times n_1}$, $u \in \mathbb{R}_+^{n_1 \times d}$ are the parameters of the network which we optimize. The function class in (1) can be seen as a generalized polynomial in the sense that the powers do not have to be integers. Polynomial neural networks have been recently analyzed in [15]. Please note that a ReLU activation function makes no sense in our setting as we require the data as well as the weights to be nonnegative. Even though nonnegativity of the weights is a strong constraint, one can model quite complex decision boundaries (see Figure 1, where we show the outcome of our method for a toy dataset in $\mathbb{R}^2$).

In order to simplify the notation we use $w = (w_1, \ldots, w_K)$ for the $K$ output units $w_i \in \mathbb{R}_+^{n_1}$, $i = 1, \ldots, K$. All output units and the hidden layer are normalized. We optimize over the set

$$S_+ = \Big\{ (w, u) \in \mathbb{R}_+^{K \times n_1} \times \mathbb{R}_+^{n_1 \times d} \ \Big| \ \|u\|_{p_u} = \rho_u, \ \|w_i\|_{p_w} = \rho_w, \ \forall i = 1, \ldots, K \Big\}.$$

We also introduce $S_{++}$ where one replaces $\mathbb{R}_+$ with $\mathbb{R}_{++} = \{t \in \mathbb{R} \mid t > 0\}$. The final optimization problem we are going to solve is given as

$$\max_{(w,u) \in S_+} \Phi(w, u) \quad \text{with} \tag{2}$$

$$\Phi(w, u) = \frac{1}{n} \sum_{i=1}^{n} \Big[ -L\big(y_i, f(w,u)(x^i)\big) + \sum_{r=1}^{K} f_r(w,u)(x^i) \Big] + \epsilon\Big( \sum_{r=1}^{K} \sum_{l=1}^{n_1} w_{r,l} + \sum_{l=1}^{n_1} \sum_{m=1}^{d} u_{lm} \Big),$$

where $(x^i, y_i) \in \mathbb{R}_+^d \times [K]$, $i = 1, \ldots, n$ is the training data. Note that this is a maximization problem and thus we use minus the loss in the objective so that we are effectively minimizing the loss. The reason to write this as a maximization problem is that our nonlinear spectral method is inspired by the theory of (sub)-homogeneous nonlinear eigenproblems on convex cones [14] which has its origin in the Perron-Frobenius theory for nonnegative matrices. In fact our work is motivated by the closely related Perron-Frobenius theory for multi-homogeneous problems developed in [7]. This is also the reason why we have nonnegative weights, as we work on the positive orthant which is a convex cone. Note that $\epsilon > 0$ in the objective can be chosen arbitrarily small and is added out of technical reasons.

In order to state our main theorem we need some additional notation. For $p \in (1, \infty)$, we let $p' = p/(p-1)$ be the Hölder conjugate of $p$, and $\psi_p(x) = \text{sign}(x)|x|^{p-1}$. We apply $\psi_p$ to scalars and vectors in which case the function is applied componentwise. For a square matrix $A$ we denote its spectral radius by $\rho(A)$. Finally, we write $\nabla_{w_i}\Phi(w, u)$ (resp. $\nabla_u\Phi(w, u)$) to denote the gradient of $\Phi$ with respect to $w_i$ (resp. $u$) at $(w, u)$. The mapping

$$G^{\Phi}(w, u) = \left( \frac{\rho_w \psi_{p'_w}(\nabla_{w_1}\Phi(w, u))}{\|\psi_{p'_w}(\nabla_{w_1}\Phi(w, u))\|_{p_w}}, \ldots, \frac{\rho_w \psi_{p'_w}(\nabla_{w_K}\Phi(w, u))}{\|\psi_{p'_w}(\nabla_{w_K}\Phi(w, u))\|_{p_w}}, \frac{\rho_u \psi_{p'_u}(\nabla_u\Phi(w, u))}{\|\psi_{p'_u}(\nabla_u\Phi(w, u))\|_{p_u}} \right),$$

(3)

defines a sequence converging to the global optimum of (2). Indeed, we prove:

**Theorem 1.** *Let $\{x^i, y_i\}_{i=1}^n \subset \mathbb{R}_+^d \times [K]$, $p_w, p_u \in (1, \infty)$, $\rho_w, \rho_u > 0$, $n_1 \in \mathbb{N}$ and $\alpha \in \mathbb{R}^{n_1}$ with $\alpha_i \geq 1$ for every $i \in [n_1]$. Define $\rho_x, \xi_1, \xi_2 > 0$ as $\rho_x = \max_{i \in [n]} \|x^i\|_1$, $\xi_1 = \rho_w \sum_{l=1}^{n_1}(\rho_u \rho_x)^{\alpha_l}$, $\xi_2 = \rho_w \sum_{l=1}^{n_1} \alpha_l (\rho_u \rho_x)^{\alpha_l}$ and let $A \in \mathbb{R}_{++}^{(K+1) \times (K+1)}$ be defined as*

$$A_{l,m} = 4(p'_w - 1)\xi_1, \qquad A_{l,K+1} = 2(p'_w - 1)(2\xi_2 + \|\alpha\|_\infty), \qquad \forall m, l \in [K].$$
$$A_{K+1,m} = 2(p'_u - 1)(2\xi_1 + 1), \quad A_{K+1,K+1} = 2(p'_u - 1)(2\xi_2 + \|\alpha\|_\infty - 1),$$

*If the spectral radius $\rho(A)$ of $A$ satisfies $\rho(A) < 1$, then (2) has a unique global maximizer $(w^*, u^*) \in S_{++}$. Moreover, for every $(w^0, u^0) \in S_{++}$, there exists $R > 0$ such that*

$$\lim_{k \to \infty} (w^k, u^k) = (w^*, u^*) \qquad and \qquad \|(w^k, u^k) - (w^*, u^*)\|_\infty \leq R\,\rho(A)^k \qquad \forall k \in \mathbb{N},$$

*where $(w^{k+1}, u^{k+1}) = G^{\Phi}(w^k, u^k)$ for every $k \in \mathbb{N}$.*

Note that one can check for a given model (number of hidden units $n_1$, choice of $\alpha$, $p_w$, $p_u$, $\rho_u$, $\rho_w$) easily if the convergence guarantee to the global optimum holds by computing the spectral radius of a square matrix of size $K + 1$. As our bounds for the matrix $A$ are very conservative, the "effective" spectral radius is typically much smaller, so that we have very fast convergence in only a few iterations, see Section 5 for a discussion. Up to our knowledge this is the first practically feasible algorithm to achieve global optimality for a non-trivial neural network model. Additionally, compared to stochastic gradient descent, there is no free parameter in the algorithm. Thus no careful tuning of the learning rate is required. The reader might wonder why we add the second term in the objective, where we sum over all outputs. The reason is that we need that the gradient of $G^{\Phi}$ is strictly positive in $S_+$, this is why we also have to add the third term for arbitrarily small $\epsilon > 0$. In Section 5 we show that this model achieves competitive results on a few UCI datasets.

**Choice of $\alpha$:** It turns out that in order to get a non-trivial classifier one has to choose $\alpha_1, \ldots, \alpha_{n_1} \geq 1$ so that $\alpha_i \neq \alpha_j$ for every $i, j \in [n_1]$ with $i \neq j$. The reason for this lies in certain invariance properties of the network. Suppose that we use a permutation invariant componentwise activation function $\sigma$, that is $\sigma(Px) = P\sigma(x)$ for any permutation matrix $P$ and suppose that $A, B$ are globally optimal weight matrices for a one hidden layer architecture, then for any permutation matrix $P$,

$$A\sigma(Bx) = AP^T P\sigma(Bx) = AP^T \sigma(PBx),$$

which implies that $A' = AP^T$ and $B' = PB$ yield the same function and thus are also globally optimal. In our setting we know that the global optimum is *unique* and thus it has to hold that, $A = AP^T$ and $B = PB$ for all permutation matrices $P$. This implies that both $A$ and $B$ have rank one and thus lead to trivial classifiers. This is the reason why one has to use different $\alpha$ for every unit.

**Dependence of $\rho(A)$ on the model parameters:** Let $Q, \tilde{Q} \in \mathbb{R}_+^{m \times m}$ and assume $0 \leq Q_{i,j} \leq \tilde{Q}_{i,j}$ for every $i,j \in [m]$, then $\rho(Q) \leq \rho(\tilde{Q})$, see Corollary 3.30 [3]. It follows that $\rho(A)$ in Theorem 1 is increasing w.r.t. $\rho_u, \rho_w, \rho_x$ and the number of hidden units $n_1$. Moreover, $\rho(A)$ is decreasing w.r.t. $p_u, p_w$ and in particular, we note that for any fixed architecture $(n_1, \alpha, \rho_u, \rho_w)$ it is always possible to find $p_u, p_w$ large enough so that $\rho(A) < 1$. Indeed, we know from the Collatz-Wielandt formula (Theorem 8.1.26 in [10]) that $\rho(A) = \rho(A^T) \leq \max_{i \in [K+1]} (A^T v)_i / v_i$ for any $v \in \mathbb{R}_{++}^{K+1}$. We use this to derive lower bounds on $p_u, p_w$ that ensure $\rho(A) < 1$. Let $v = (p_w - 1, \ldots, p_w - 1, p_u - 1)$, then $(A^T v)_i < v_i$ for every $i \in [K+1]$ guarantees $\rho(A) < 1$ and is equivalent to

$$p_w > 4(K+1)\xi_1 + 3 \qquad \text{and} \qquad p_u > 2(K+1)(\|\alpha\|_\infty + 2\xi_2) - 1, \qquad (4)$$

where $\xi_1, \xi_2$ are defined as in Theorem 1. However, we think that our current bounds are sub-optimal so that this choice is quite conservative. Finally, we note that the constant $R$ in Theorem 1 can be explicitly computed when running the algorithm (see Theorem 3).

**Proof Strategy:** The following main part of the paper is devoted to the proof of the algorithm. For that we need some further notation. We introduce the sets

$$V_+ = \mathbb{R}_+^{K \times n_1} \times \mathbb{R}_+^{n_1 \times d}, \quad V_{++} = \mathbb{R}_{++}^{K \times n_1} \times \mathbb{R}_{++}^{n_1 \times d}$$

$$B_+ = \left\{ (w,u) \in V_+ \mid \|u\|_{p_u} \leq \rho_u, \; \|w_i\|_{p_w} \leq \rho_w, \; \forall i = 1, \ldots, K \right\},$$

and similarly we define $B_{++}$ replacing $V_+$ by $V_{++}$ in the definition. The high-level idea of the proof is that we first show that the global maximum of our optimization problem in (2) is attained in the "interior" of $S_+$, that is $S_{++}$. Moreover, we prove that any critical point of (2) in $S_{++}$ is a fixed point of the mapping $G^\Phi$. Then we proceed to show that there exists a unique fixed point of $G^\Phi$ in $S_{++}$ and thus there is a unique critical point of (2) in $S_{++}$. As the global maximizer of (2) exists and is attained in the interior, this fixed point has to be the global maximizer.

Finally, the proof of the fact that $G^\Phi$ has a unique fixed point follows by noting that $G^\Phi$ maps $B_{++}$ into $B_{++}$ and the fact that $B_{++}$ is a complete metric space with respect to the Thompson metric. We provide a characterization of the Lipschitz constant of $G^\Phi$ and in turn derive conditions under which $G^\Phi$ is a contraction. Finally, the application of the Banach fixed point theorem yields the uniqueness of the fixed point of $G^\Phi$ and the linear convergence rate to the global optimum of (2). In Section 4 we show the application of the established framework for our neural networks.

## 3 From the optimization problem to fixed point theory

**Lemma 1.** *Let $\Phi : V \to \mathbb{R}$ be differentiable. If $\nabla\Phi(w,u) \in V_{++}$ for every $(w,u) \in S_+$, then the global maximum of $\Phi$ on $S_+$ is attained in $S_{++}$.*

We now identify critical points of the objective $\Phi$ in $S_{++}$ with fixed points of $G^\Phi$ in $S_{++}$.

**Lemma 2.** *Let $\Phi : V \to \mathbb{R}$ be differentiable. If $\nabla\Phi(w,u) \in V_{++}$ for all $(w,u) \in S_{++}$, then $(w^*, u^*)$ is a critical point of $\Phi$ in $S_{++}$ if and only if it is a fixed point of $G^\Phi$.*

Our goal is to apply the Banach fixed point theorem to $G^\Phi : B_{++} \to S_{++} \subset B_{++}$. We recall this theorem for the convenience of the reader.

**Theorem 2** (Banach fixed point theorem e.g. [12])**.** *Let $(X, d)$ be a complete metric space with a mapping $T : X \to X$ such that $d(T(x), T(y)) \leq q\, d(x,y)$ for $q \in [0,1)$ and all $x, y \in X$. Then $T$ has a unique fixed-point $x^*$ in $X$, that is $T(x^*) = x^*$ and the sequence defined as $x^{n+1} = T(x^n)$ with $x^0 \in X$ converges $\lim_{n \to \infty} x^n = x^*$ with linear convergence rate*

$$d(x^n, x^*) \; \leq \; \frac{q^n}{1-q} \, d(x^1, x^0).$$

So, we need to endow $B_{++}$ with a metric $\mu$ so that $(B_{++}, \mu)$ is a complete metric space. A popular metric for the study of nonlinear eigenvalue problems on the positive orthant is the so-called Thompson metric $d : \mathbb{R}_{++}^m \times \mathbb{R}_{++}^m \to \mathbb{R}_+$ [18] defined as

$$d(z, \tilde{z}) = \|\ln(z) - \ln(\tilde{z})\|_\infty \qquad \text{where} \qquad \ln(z) = \big(\ln(z_1), \ldots, \ln(z_m)\big).$$

Using the known facts that $(\mathbb{R}^n_{++}, d)$ is a complete metric space and its topology coincides with the norm topology (see e.g. Corollary 2.5.6 and Proposition 2.5.2 [14]), we prove:

**Lemma 3.** *For $p \in (1, \infty)$ and $\rho > 0$, $(\{z \in \mathbb{R}^n_{++} \mid \|z\|_p \leq \rho\}, d)$ is a complete metric space.*

Now, the idea is to see $B_{++}$ as a product of such metric spaces. For $i = 1, \ldots, K$, let $B^i_{++} = \{w_i \in \mathbb{R}^{n_1}_{++} \mid \|w_i\|_{p_w} \leq \rho_w\}$ and $d_i(w_i, \tilde{w}_i) = \gamma_i \|\ln(w_i) - \ln(\tilde{w}_i)\|_\infty$ for some constant $\gamma_i > 0$. Furthermore, let $B^{K+1}_{++} = \{u \in \mathbb{R}^{n_1 \times d}_{++} \mid \|u\|_{p_u} \leq \rho_u\}$ and $d_{K+1}(u, \tilde{u}) = \gamma_{K+1} \|\ln(u) - \ln(\tilde{u})\|_\infty$. Then $(B^i_{++}, d_i)$ is a complete metric space for every $i \in [K+1]$ and $B_{++} = B^1_{++} \times \ldots \times B^K_{++} \times B^{K+1}_{++}$. It follows that $(B_{++}, \mu)$ is a complete metric space with $\mu \colon B_{++} \times B_{++} \to \mathbb{R}_+$ defined as

$$\mu\big((w, u), (\tilde{w}, \tilde{u})\big) = \sum_{i=1}^K \gamma_i \|\ln(w_i) - \ln(\tilde{w}_i)\|_\infty + \gamma_{K+1} \|\ln(u) - \ln(\tilde{u})\|_\infty.$$

The motivation for introducing the weights $\gamma_1, \ldots, \gamma_{K+1} > 0$ is given by the next theorem. We provide a characterization of the Lipschitz constant of a mapping $F \colon B_{++} \to B_{++}$ with respect to $\mu$. Moreover, this Lipschitz constant can be minimized by a smart choice of $\gamma$. For $i \in [K], a, j \in [n_1], b \in [d]$, we write $F_{w_{i,j}}$ and $F_{u_{ab}}$ to denote the components of $F$ such that $F = (F_{w_{1,1}}, \ldots, F_{w_{1,n_1}}, F_{w_{2,1}}, \ldots, F_{w_{K,n_1}}, F_{u_{11}}, \ldots, F_{u_{n_1 d}})$.

**Lemma 4.** *Suppose that $F \in C^1(B_{++}, V_{++})$ and $A \in \mathbb{R}^{(K+1) \times (K+1)}_+$ satisfies*

$$\big\langle |\nabla_{w_k} F_{w_{i,j}}(w, u)|, w_k \big\rangle \leq A_{i,k} \, F_{w_{i,j}}(w, u), \quad \big\langle |\nabla_u F_{w_{i,j}}(w, u)|, u \big\rangle \leq A_{i,K+1} \, F_{w_{i,j}}(w, u)$$

*and*

$$\big\langle |\nabla_{w_k} F_{u_{ab}}(w, u)|, w_k \big\rangle \leq A_{K+1,k} \, F_{u_{ab}}(w, u), \quad \big\langle |\nabla_u F_{u_{ab}}(w, u)|, u \big\rangle \leq A_{K+1,K+1} \, F_{u_{ab}}(w, u)$$

*for all $i, k \in [K]$, $a, j \in [n_1]$, $b \in [d]$ and $(w, u) \in B_{++}$. Then, for every $(w, u), (\tilde{w}, \tilde{u}) \in B_{++}$ it holds*

$$\mu\big(F(w, u), F(\tilde{w}, \tilde{u})\big) \leq U \, \mu\big((w, u), (\tilde{w}, \tilde{u})\big) \qquad with \qquad U = \max_{k \in [K+1]} \frac{(A^T \gamma)_k}{\gamma_k}.$$

Note that, from the Collatz-Wielandt ratio for nonnegative matrices, we know that the constant $U$ in Lemma 4 is lower bounded by the spectral radius $\rho(A)$ of $A$. Indeed, by Theorem 8.1.31 in [10], we know that if $A^T$ has a positive eigenvector $\gamma \in \mathbb{R}^{K+1}_{++}$, then

$$\max_{i \in [K+1]} \frac{(A^T \gamma)_i}{\gamma_i} = \rho(A) = \min_{\tilde{\gamma} \in \mathbb{R}^{K+1}_{++}} \max_{i \in [K+1]} \frac{(A^T \tilde{\gamma})_i}{\tilde{\gamma}_i}. \tag{5}$$

Therefore, in order to obtain the minimal Lipschitz constant $U$ in Lemma 4, we choose the weights of the metric $\mu$ to be the components of $\gamma$. A combination of Theorem 2, Lemma 4 and this observation implies the following result.

**Theorem 3.** *Let $\Phi \in C^1(V, \mathbb{R}) \cap C^2(B_{++}, \mathbb{R})$ with $\nabla \Phi(S_+) \subset V_{++}$. Let $G^\Phi \colon B_{++} \to B_{++}$ be defined as in (3). Suppose that there exists a matrix $A \in \mathbb{R}^{(K+1) \times (K+1)}_+$ such that $G^\Phi$ and $A$ satisfies the assumptions of Lemma 4 and $A^T$ has a positive eigenvector $\gamma \in \mathbb{R}^{K+1}_{++}$. If $\rho(A) < 1$, then $\Phi$ has a unique critical point $(w^*, u^*)$ in $S_{++}$ which is the global maximum of the optimization problem (2). Moreover, the sequence $\big((w^k, u^k)\big)_k$ defined for any $(w^0, u^0) \in S_{++}$ as $(w^{k+1}, u^{k+1}) = G^\Phi(w^k, u^k)$, $k \in \mathbb{N}$, satisfies $\lim_{k \to \infty} (w^k, u^k) = (w^*, u^*)$ and*

$$\|(w^k, u^k) - (w^*, u^*)\|_\infty \leq \rho(A)^k \left( \frac{\mu\big((w^1, u^1), (w^0, u^0)\big)}{\big(1 - \rho(A)\big) \min\big\{\frac{\gamma_{K+1}}{\rho_u}, \min_{t \in [K]} \frac{\gamma_t}{\rho_w}\big\}} \right) \qquad \forall k \in \mathbb{N},$$

*where the weights in the definition of $\mu$ are the entries of $\gamma$.*

## 4 Application to Neural Networks

In the previous sections we have outlined the proof of our main result for a general objective function satisfying certain properties. The purpose of this section is to prove that the properties hold for our optimization problem for neural networks.

We recall our objective function from (2)

$$\Phi(w,u) = \frac{1}{n}\sum_{i=1}^{n}\Big[-L\big(y_i, f(w,u)(x^i)\big) + \sum_{r=1}^{K} f_r(w,u)(x^i)\Big] + \epsilon\Big(\sum_{r=1}^{K}\sum_{l=1}^{n_1} w_{r,l} + \sum_{l=1}^{n_1}\sum_{m=1}^{d} u_{lm}\Big)$$

and the function class we are considering from (1)

$$f_r(x) = f_r(w,u)(x) = \sum_{l=1}^{n_1} w_{r,l}\Big(\sum_{m=1}^{d} u_{lm}x_m\Big)^{\alpha_l},$$

The arbitrarily small $\epsilon$ in the objective is needed to make the gradient strictly positive on the boundary of $V_+$. We note that the assumption $\alpha_i \geq 1$ for every $i \in [n_1]$ is crucial in the following lemma in order to guarantee that $\nabla\Phi$ is well defined on $S_+$.

**Lemma 5.** *Let $\Phi$ be defined as in (2), then $\nabla\Phi(w,u)$ is strictly positive for any $(w,u) \in S_+$.*

Next, we derive the matrix $A \in \mathbb{R}^{(K+1)\times(K+1)}$ in order to apply Theorem 3 to $G^\Phi$ with $\Phi$ defined in (2). As discussed in its proof, the matrix $A$ given in the following theorem has a smaller spectral radius than that of Theorem 1. To express this matrix, we consider $\Psi_{p,q}^\alpha : \mathbb{R}_{++}^{n_1} \times \mathbb{R}_{++} \to \mathbb{R}_{++}$ defined for $p,q \in (1,\infty)$ and $\alpha \in \mathbb{R}_{++}^{n_1}$ as

$$\Psi_{p,q}^\alpha(\delta,t) = \left(\Big[\sum_{l\in J}(\delta_l\,t^{\alpha_l})^{\frac{p\,q}{q-\overline{\alpha}p}}\Big]^{1-\frac{\overline{\alpha}p}{q}} + \max_{j\in J^c}(\delta_j\,t^{\alpha_j})^p\right)^{1/p}, \tag{6}$$

where $J = \{l \in [n_1] \mid \alpha_l p \leq q\}$, $J^c = \{l \in [n_1] \mid \alpha_l p > q\}$ and $\overline{\alpha} = \min_{l\in J}\alpha_l$.

**Theorem 4.** *Let $\Phi$ be defined as above and $G^\Phi$ be as in (3). Set $C_w = \rho_w\,\Psi_{p'_w,p_u}^\alpha(\mathbf{1},\rho_u\rho_x)$, $C_u = \rho_w\,\Psi_{p'_w,p_u}^\alpha(\alpha,\rho_u\rho_x)$ and $\rho_x = \max_{i\in[n]}\|x^i\|_{p'_u}$. Then $A$ and $G^\Phi$ satisfy all assumptions of Lemma 4 with*

$$A = 2\operatorname{diag}\big(p'_w-1,\ldots,p'_w-1,p'_u-1\big)\begin{pmatrix} Q_{w,w} & Q_{w,u} \\ Q_{u,w} & Q_{u,u} \end{pmatrix}$$

*where $Q_{w,w} \in \mathbb{R}_{++}^{K\times K}, Q_{w,u} \in \mathbb{R}_{++}^{K\times 1}, Q_{u,w} \in \mathbb{R}_{++}^{1\times K}$ and $Q_{u,u} \in \mathbb{R}_{++}$ are defined as*

$$\begin{aligned} Q_{w,w} &= 2C_w\mathbf{1}\mathbf{1}^T, & Q_{w,u} &= (2C_u + \|\alpha\|_\infty)\mathbf{1}, \\ Q_{u,w} &= (2C_w+1)\mathbf{1}^T, & Q_{u,u} &= (2C_u + \|\alpha\|_\infty - 1). \end{aligned}$$

In the supplementary material, we prove that $\Psi_{p,q}^\alpha(\delta,t) \leq \sum_{l=1}^{n_1}\delta_l t^{\alpha_l}$ which yields the weaker bounds $\xi_1, \xi_2$ given in Theorem 1. In particular, this observation combined with Theorems 3 and 4 implies Theorem 1.

## 4.1  Neural networks with two hidden layers

We show how to extend our framework for neural networks with 2 hidden layers. In future work we will consider the general case. We briefly explain the major changes. Let $n_1, n_2 \in \mathbb{N}$ and $\alpha \in \mathbb{R}_{++}^{n_1}, \beta \in \mathbb{R}_{++}^{n_2}$ with $\alpha_i, \beta_j \geq 1$ for all $i \in [n_1], j \in [n_2]$, our function class is:

$$f_r(x) = f_r(w,v,u)(x) = \sum_{l=1}^{n_2} w_{r,l}\Big(\sum_{m=1}^{n_1} v_{lm}\Big(\sum_{s=1}^{d} u_{ms}x_s\Big)^{\alpha_m}\Big)^{\beta_l}$$

and the optimization problem becomes

$$\max_{(w,v,u)\in S_+}\Phi(w,v,u) \qquad \text{where} \qquad V_+ = \mathbb{R}_+^{K\times n_2} \times \mathbb{R}_+^{n_2\times n_1} \times \mathbb{R}_+^{n_1\times d}, \tag{7}$$

$S_+ = \{(w_1,\ldots,w_K,v,u) \in V_+ \mid \|w_i\|_{p_w} = \rho_w, \|v\|_{p_v} = \rho_v, \|u\|_{p_u} = \rho_u\}$ and

$$\Phi(w,v,u) = \frac{1}{n}\sum_{i=1}^{n}\Big[-L\big(y_i, f(x^i)\big) + \sum_{r=1}^{K} f_r(x^i)\Big] + \epsilon\Big(\sum_{r=1}^{K}\sum_{l=1}^{n_2} w_{r,l} + \sum_{l=1}^{n_2}\sum_{m=1}^{n_1} v_{lm} + \sum_{m=1}^{n_1}\sum_{s=1}^{d} u_{ms}\Big).$$

The map $G^\Phi\colon S_{++} \to S_{++} = \{z \in S_+ \mid z > 0\}$, $G^\Phi = (G^\Phi_{w_1}, \dots, G^\Phi_{w_K}, G^\Phi_v, G^\Phi_u)$, becomes

$$G^\Phi_{w_i}(w, v, u) = \rho_w \frac{\psi_{p'_w}(\nabla_{w_i}\Phi(w, u))}{\|\psi_{p'_w}(\nabla_{w_i}\Phi(w, v, u))\|_{p_w}} \qquad \forall i \in [K] \tag{8}$$

and

$$G^\Phi_v(w, v, u) = \rho_v \frac{\psi_{p'_v}(\nabla_v\Phi(w, v, u))}{\|\psi_{p'_v}(\nabla_v\Phi(w, v, u))\|_{p_v}}, \qquad G^\Phi_u(w, v, u) = \rho_u \frac{\psi_{p'_u}(\nabla_u\Phi(w, v, u))}{\|\psi_{p'_u}(\nabla_u\Phi(w, v, u))\|_{p_u}}.$$

We have the following equivalent of Theorem 1 for 2 hidden layers.

**Theorem 5.** *Let* $\{x^i, y_i\}_{i=1}^n \subset \mathbb{R}^d_+ \times [K]$, $p_w, p_v, p_u \in (1, \infty)$, $\rho_w, \rho_v, \rho_u > 0$, $n_1, n_2 \in \mathbb{N}$ *and* $\alpha \in \mathbb{R}^{n_1}_{++}, \beta \in \mathbb{R}^{n_2}_{++}$ *with* $\alpha_i, \beta_j \geq 1$ *for all* $i \in [n_1], j \in [n_2]$. *Let* $\rho_x = \max_{i \in [n]} \|x^i\|_{p'_u}$,

$$\theta = \rho_v \Psi^\alpha_{p'_v, p_u}(\mathbf{1}, \rho_u \rho_x), \quad C_w = \rho_w \Psi^\beta_{p'_w, p_v}(\mathbf{1}, \theta), \quad C_v = \rho_w \Psi^\beta_{p'_w, p_v}(\beta, \theta), \quad C_u = \|\alpha\|_\infty C_v,$$

*and define* $A \in \mathbb{R}^{(K+2)\times(K+2)}_{++}$ *as*

$$
\begin{aligned}
A_{m,l} &= 4(p'_w - 1)C_w, & A_{m,K+1} &= 2(p'_w - 1)(2C_v + \|\beta\|_\infty) \\
A_{m,K+2} &= 2(p'_w - 1)\big(2C_u + \|\alpha\|_\infty\|\beta\|_\infty\big), & A_{K+1,l} &= 2(p'_v - 1)\big(2C_w + 1\big) \\
A_{K+1,K+1} &= 2(p'_v - 1)\big(2C_v + \|\beta\|_\infty - 1\big), & A_{K+1,K+2} &= 2(p'_v - 1)\big(2C_u + \|\alpha\|_\infty\|\beta\|_\infty\big) \\
A_{K+2,l} &= 2(p'_u - 1)(2C_w + 1), & A_{K+2,K+1} &= 2(p'_u - 1)(2C_v + \|\beta\|_\infty), \\
A_{K+2,K+2} &= 2(p'_u - 1)(2C_u + \|\alpha\|_\infty\|\beta\|_\infty - 1) & &\forall m, l \in [K].
\end{aligned}
$$

*If* $\rho(A) < 1$, *then* (7) *has a unique global maximizer* $(w^*, v^*, u^*) \in S_{++}$. *Moreover, for every* $(w^0, v^0, u^0) \in S_{++}$, *there exists* $R > 0$ *such that*

$$\lim_{k \to \infty}(w^k, v^k, u^k) = (w^*, v^*, u^*) \qquad and \qquad \|(w^k, v^k, u^k) - (w^*, v^*, u^*)\|_\infty \leq R\,\rho(A)^k \quad \forall k \in \mathbb{N}$$

*where* $(w^{k+1}, v^{k+1}, u^{k+1}) = G^\Phi(w^k, v^k, u^k)$ *for every* $k \in \mathbb{N}$ *and* $G^\Phi$ *is defined as in* (8).

As for the case with one hidden layer, for any fixed architecture $\rho_w, \rho_v, \rho_u > 0$, $n_1, n_2 \in \mathbb{N}$ and $\alpha \in \mathbb{R}^{n_1}_{++}, \beta \in \mathbb{R}^{n_2}_{++}$ with $\alpha_i, \beta_j \geq 1$ for all $i \in [n_1], j \in [n_2]$, it is possible to derive lower bounds on $p_w, p_v, p_u$ that guarantee $\rho(A) < 1$ in Theorem 5. Indeed, it holds

$$C_w \leq \zeta_1 = \rho_w \sum_{j=1}^{n_2}\left[\rho_v \sum_{l=1}^{n_1}(\rho_u\tilde{\rho}_x)^{\alpha_l}\right]^{\beta_j} \quad \text{and} \quad C_v \leq \zeta_2 = \rho_w \sum_{j=1}^{n_2}\beta_j\left[\rho_v \sum_{l=1}^{n_1}(\rho_u\tilde{\rho}_x)^{\alpha_l}\right]^{\beta_j},$$

with $\tilde{\rho}_x = \max_{i \in [n]}\|x^i\|_1$. Hence, the two hidden layers equivalent of (4) becomes

$$p_w > 4(K+2)\zeta_1 + 5, \; p_v > 2(K+2)\big[2\zeta_2 + \|\beta\|_\infty\big] - 1, \; p_u > 2(K+2)\|\alpha\|_\infty(2\zeta_2 + \|\beta\|_\infty) - 1. \tag{9}$$

## 5 Experiments

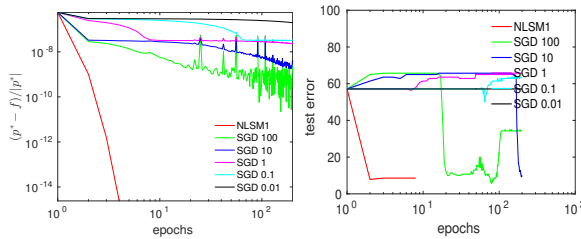

**Figure 2:** Training score (left) w.r.t. the optimal score $p^*$ and test error (right) of NLSM1 and Batch-SGD with different step-sizes.

**Table 1:** Test accuracy on UCI datasets

| Dataset | NLSM1 | NLSM2 | ReLU1 | ReLU2 | SVM |
|---|---|---|---|---|---|
| Cancer | 96.4 | 96.4 | 95.7 | 93.6 | 95.7 |
| Iris | 90.0 | 96.7 | 100 | 93.3 | 100 |
| Banknote | 97.1 | 96.4 | 100 | 97.8 | 100 |
| Blood | 76.0 | 76.7 | 76.0 | 76.0 | 77.3 |
| Haberman | 75.4 | 75.4 | 70.5 | 72.1 | 72.1 |
| Seeds | 88.1 | 90.5 | 90.5 | 92.9 | 95.2 |
| Pima | 79.2 | 80.5 | 76.6 | 79.2 | 79.9 |

The shown experiments should be seen as a proof of concept. We do not have yet a good understanding of how one should pick the parameters of our model to achieve good performance. However, the other papers which have up to now discussed global optimality for neural networks [11, 8] have not included any results on real datasets. Thus, up to our

<div align="center">Nonlinear Spectral Method for 1 hidden layer</div>

| |
|---|
| **Input:** Model $n_1 \in \mathbb{N}$, $p_w, p_u \in (1, \infty)$, $\rho_w, \rho_u > 0$, $\alpha_1, \ldots, \alpha_{n_1} \geq 1$, $\epsilon > 0$ so that the matrix $A$ of Theorem 1 satisfies $\rho(A) < 1$. Accuracy $\tau > 0$ and $(w^0, u^0) \in S_{++}$. |
| 1    Let $(w^1, u^1) = G^\Phi(w^0, u^0)$ and compute $R$ as in Theorem 3 |
| 2    **Repeat** |
| 3        $(w^{k+1}, u^{k+1}) = G^\Phi(w^k, u^k)$ |
| 4        $k \leftarrow k + 1$ |
| 5    **Until** $k \geq \ln\big(\tau/R\big)/\ln\big(\rho(A)\big)$ |
| **Output:** $(w^k, u^k)$ fulfills $\|(w^k, u^k) - (w^*, u^*)\|_\infty < \tau$. |

<div align="center">With $G^\Phi$ defined as in (3). The method for two hidden layers is similar: consider $G^\Phi$<br>as in (8) instead of (3) and assume that the model satisfies Theorem 5.</div>

knowledge, we show for the first time a globally optimal algorithm for neural networks that leads to non-trivial classification results.

We test our methods on several low dimensional UCI datasets and denote our algorithms as NLSM1 (one hidden layer) and NLSM2 (two hidden layers). We choose the parameters of our model out of 100 randomly generated combinations of $(n_1, \alpha, \rho_w, \rho_u) \in [2, 20] \times [1, 4] \times (0, 1]^2$ (respectively $(n_1, n_2, \alpha, \beta, \rho_w, \rho_v, \rho_u) \in [2, 10]^2 \times [1, 4]^2 \times (0, 1]^2$) and pick the best one based on 5-fold cross-validation error. We use Equation (4) (resp. Equation (9)) to choose $p_u, p_w$ (resp. $p_u, p_v, p_w$) so that every generated model satisfies the conditions of Theorem 1 (resp. Theorem 5), i.e. $\rho(A) < 1$. Thus, global optimality is guaranteed in all our experiments. For comparison, we use the nonlinear RBF-kernel SVM and implement two versions of the Rectified-Linear Unit network - one for one hidden layer networks (ReLU1) and one for two hidden layers networks (ReLU2). To train ReLU, we use a stochastic gradient descent method which minimizes the sum of logistic loss and $L_2$ regularization term over weight matrices to avoid over-fitting. All parameters of each method are jointly cross validated. More precisely, for ReLU the number of hidden units takes values from 2 to 20, the step-sizes and regularizers are taken in $\{10^{-6}, 10^{-5}, \ldots, 10^2\}$ and $\{0, 10^{-4}, 10^{-3}, \ldots, 10^4\}$ respectively. For SVM, the hyperparameter $C$ and the kernel parameter $\gamma$ of the radius basis function $K(x^i, x^j) = \exp(-\gamma\|x^i - x^j\|^2)$ are taken from $\{2^{-5}, 2^{-4} \ldots, 2^{20}\}$ and $\{2^{-15}, 2^{-14} \ldots, 2^3\}$ respectively. Note that ReLUs allow negative weights while our models do not. The results presented in Table 1 show that overall our nonlinear spectral methods achieve slightly worse performance than kernel SVM while being competitive/slightly better than ReLU networks. Notably in case of Cancer, Haberman and Pima, NLSM2 outperforms all the other models. For Iris and Banknote, we note that without any constraints ReLU1 can easily find an architecture which achieves zero test error while this is difficult for our models as we impose constraints on the architecture in order to prove global optimality.

We compare our algorithms with Batch-SGD in order to optimize (2) with batch-size being 5% of the training data while the step-size is fixed and selected between $10^{-2}$ and $10^2$. At each iteration of our spectral method and each epoch of Batch-SGD, we compute the objective and test error of each method and show the results in Figure 2. One can see that our method is much faster than SGDs, and has a linear convergence rate. We noted in our experiments that as $\alpha$ is large and our data lies between $[0, 1]$, all units in the network tend to have small values that make the whole objective function relatively small. Thus, a relatively large change in $(w, u)$ might cause only small changes in the objective function but performance may vary significantly as the distance is large in the parameter space. In other words, a small change in the objective may have been caused by a large change in the parameter space, and thus, largely influences the performance - which explains the behavior of SGDs in Figure 2.

The magnitude of the entries of the matrix $A$ in Theorems 1 and 5 grows with the number of hidden units and thus the spectral radius $\rho(A)$ also increases with this number. As we expect that the number of required hidden units grows with the dimension of the datasets we have limited ourselves in the experiments to low-dimensional datasets. However, these bounds are likely not to be tight, so that there might be room for improvement in terms of dependency on the number of hidden units.

<div align="center">8</div>

## Acknowledgment

The authors acknowledge support by the ERC starting grant NOLEPRO 307793.

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
