[Supplementary Material]

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

*Proof.* First note that as $\Phi$ is a continuous function on the compact set $S_+$ the global minimum and maximum are attained. A boundary point $(w^b, u^b)$ of $S_+$ is characterized by the fact that at least one of the variables $(w^b, u^b)$ has a zero component. Suppose w.l.o.g. that the subset $J \subset [n_1]$ of components of $w_1^b \in \mathbb{R}^{n_1}_+$ are zero, that is $w_{1,J}^b = 0$. The normal vector of the $p_w$-sphere at $w_1^b$ is given by $\nu = \psi_{p_w}(w_1^b)$. The set of tangent directions is thus given by

$$T = \{v \in \mathbb{R}^{n_1} \mid \langle \nu, v \rangle = 0\}.$$

Note that if $(w^b, u^b)$ is a local maximum, then

$$\left\langle \nabla_{w_1^b} \Phi(w^b, u^b), t \right\rangle \le 0, \quad \forall t \in T_+ = \{v \in \mathbb{R}^{n_1}_+ \mid \langle \nu, v \rangle = 0\}, \tag{5}$$

where $T_+$ is the set of "positive" tangent directions, that are pointing inside the set $\{w_1 \in \mathbb{R}^{n_1}_+ \mid \|w_1\|_{p_w} = \rho_w\}$. Otherwise there would exist a direction of ascent which leads to a feasible point. Now note that $\nu$ has non-negative components as $w_1^b \in \mathbb{R}^{n_1}_+$. Thus

$$T_+ = \{v \in \mathbb{R}^{n_1}_+ \mid v_i = 0 \text{ if } i \notin J\}.$$

However, by assumption $\nabla_{w_i^b}\Phi(w^b, u^b)$ is a vector with strictly positive components and thus (5) can never be fulfilled as $T_+$ contains only vectors with non-negative components and at least one of the components is strictly positive as $J \neq [n_1]$. Finally, as the global maximum is attained in $S_+$ and no local maximum exists at the boundary, the global maximum has to be attained in $S_{++}$. $\qquad\square$

We now identify critical points of the objective $\Phi$ in $S_{++}$ with fixed points of $G^\Phi$ in $S_{++}$.

**Lemma 2.** *Let $\Phi : V \to \mathbb{R}$ be differentiable. If $\nabla\Phi(w, u) \in V_{++}$ for all $(w, u) \in S_{++}$, then $(w^*, u^*)$ is a critical point of $\Phi$ in $S_{++}$ if and only if it is a fixed point of $G^\Phi$.*

*Proof.* The Lagrangian of $\Phi$ constrained to the unit sphere $S$ is given by

$$\mathcal{L}(w, u, \lambda) = \Phi(w, u) - \lambda_{K+1}(\|u\|_{p_u} - \rho_u) - \sum_{j=1}^{K} \lambda_i(\|w_j\|_{p_w} - \rho_w).$$

A necessary and sufficient condition [4] for $(w, u) \in S_{++}$ being a critical point of $\Phi$ is the existence of $\lambda_i$ with

$$\nabla_{w_j}\Phi(w, u) = \lambda_j \psi_{p_w}(w_j) \quad \forall j \in [K] \qquad \text{and} \qquad \nabla_u\Phi(w, u) = \lambda_{K+1}\psi_{p_u}(u). \qquad (6)$$

Note that as $w_j \in \mathbb{R}_{++}^{n_1}$ and the gradients are strictly positive in $S_{++}$ the $\lambda_i$, $i = 1, \ldots, K+1$ have to be strictly positive. Noting that $\psi_{p'}(\psi_p(x)) = x$, we get

$$\psi_{p'_w}\big(\nabla_{w_j}\Phi(w, u)\big) = \lambda_j^{p'_w-1} w_j \quad \forall j \in [K] \qquad \text{and} \qquad \psi_{p'_u}\big(\nabla_u\Phi(w, u)\big) = \lambda_{K+1}^{p'_u-1} u. \qquad (7)$$

In particular, $(w^*, u^*)$ is a critical point of $\Phi$ in $S_{++}$ if and only if it satisfies (7). Finally, note that

$$\left( \frac{\rho_w \psi_{p'_w}\big(\nabla_{w_1}\Phi(w, u)\big)}{\|\psi_{p'_w}\big(\nabla_{w_1}\Phi(w, u)\big)\|_{p_w}}, \ldots, \frac{\rho_w \psi_{p'_w}\big(\nabla_{w_K}\Phi(w, u)\big)}{\|\psi_{p'_w}\big(\nabla_{w_K}\Phi(w, u)\big)\|_{p_w}}, \frac{\rho_u \psi_{p'_u}\big(\nabla_u\Phi(w, u)\big)}{\|\psi_{p'_u}\big(\nabla_u\Phi(w, u)\big)\|_{p_u}} \right) \in S_{++},$$

as the gradient is strictly positive on $S_{++}$ and thus $G^\Phi : S_{++} \to S_{++}$ defined in (3) is well-defined and if $(w^*, u^*)$ is a critical point, then by (7) it holds $G^\Phi(w^*, u^*) = (w^*, u^*)$. On the other hand if $G^\Phi(w^*, u^*) = (w^*, u^*)$, then

$$\rho_w \frac{\psi_{p'_w}\big(\nabla_{w_j}\Phi(w^*, u^*)\big)}{\|\psi_{p'_w}\big(\nabla_{w_j}\Phi(w^*, u^*)\big)\|_{p_w}} = w_j^*, \quad j = 1, \ldots, K, \qquad \rho_u \frac{\psi_{p'_u}\big(\nabla_u\Phi(w^*, u^*)\big)}{\|\psi_{p'_u}\big(\nabla_u\Phi(w^*, u^*)\big)\|_{p_u}} = u^*$$

and thus there exists $\lambda_j = \frac{\|\psi_{p'_w}(\nabla_{w_j}\Phi(w^*, u^*))\|_{p_w}}{\rho_w}$, $j \in [K]$ and $\lambda_{K+1} = \frac{\|\psi_{p'_u}(\nabla_u\Phi(w^*, u^*))\|_{p_u}}{\rho_u}$ such that (7) holds and thus $(w^*, u^*)$ is a critical point of $\Phi$ in $S_{++}$. $\qquad\square$

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

*Proof.* Let $i \in [K]$ and $j \in [n_1]$. First, we show that for $(w, u), (\tilde{w}, \tilde{u}) \in B_{++}$, one has

$$\left| \ln\big(F_{w_{i,j}}(w, u)\big) - \ln\big(F_{w_{i,j}}(\tilde{w}, \tilde{u})\big) \right| \leq \sum_{s=1}^K A_{i,s} \|\ln(w_s) - \ln(\tilde{w}_s)\|_\infty + A_{i,K+1} \|\ln(u) - \ln(\tilde{u})\|_\infty.$$

Let $(w, u), (\tilde{w}, \tilde{u}) \in B_{++}$. If $(w, u) = (\tilde{w}, \tilde{u})$, there is nothing to prove. So, suppose that $(w, u) \neq (\tilde{w}, \tilde{u})$. Let $\hat{B} = \{\ln(v) \mid v \in B_{++}\} \subset V$ and $g \colon \hat{B} \to \mathbb{R}$ with

$$g(\overline{w}, \overline{u}) = \ln\left(F_{w_{i,j}}\big(\exp(\overline{w}, \overline{u})\big)\right) \quad \text{where} \quad \exp(\overline{w}, \overline{u}) = \big(e^{\overline{w}_{1,1}}, \ldots, e^{\overline{w}_{K,n_1}}, e^{\overline{u}_{11}}, \ldots, e^{\overline{u}_{n_1 d}}\big).$$

Note that $g$ is continuously differentiable because it is the composition of continuously differentiable mappings. Set $(v, x) = \big(\ln(w), \ln(u)\big)$ and $(y, z) = \big(\ln(\tilde{w}), \ln(\tilde{u})\big)$. By the mean value theorem, there exists $t \in (0, 1)$, such that $(\hat{w}, \hat{u}) = t(v, x) + (1 - t)(y, z)$ satisfies

$$g(v, x) - g(y, z) = \langle \nabla g(\hat{w}, \hat{u}), (v, x) - (y, z) \rangle.$$

Let $(\overline{w}, \overline{u}) = \exp(\hat{w}, \hat{u})$. It follows that

$$\ln\big(F_{w_{i,j}}(w, u)\big) - \ln\big(F_{w_{i,j}}(\tilde{w}, \tilde{u})\big) = \langle \nabla g(\hat{w}, \hat{u}), (v, x) - (y, z) \rangle$$

$$= \frac{\big\langle \big(\nabla F_{w_{i,j}}(\overline{w}, \overline{u})\big) \circ (\overline{w}, \overline{u}), (v, x) - (y, z) \big\rangle}{F_{w_{i,j}}(\overline{w}, \overline{u})}$$

where $\circ$ denotes the Hadamard product. Note that by convexity of the exponential and $\|\cdot\|_p$ we have $(\overline{w}, \overline{u}) \in B_{++}$ as

$$\|\overline{w}\|_{p_w} = \|e^{tv+(1-t)y}\|_{p_w} \leq \|te^v + (1-t)e^y\|_{p_w} \leq t\|e^v\|_{p_w} + (1-t)\|e^y\|_{p_w}$$
$$= t\|w\|_{p_w} + (1-t)\|\tilde{w}\|_{p_w} \leq \rho_w,$$

where we have used in the second step convexity of the exponential and that the $p$-norms are order preserving, $\|x\|_p \leq \|y\|_p$ if $x_i \leq z_i$ for all $i$. A similar argument shows $\|\overline{u}\|_{p_u} \leq \rho_u$. Hence

$$\left| \ln\left(F_{w_{i,j}}(w, u)\right) - \ln\left(F_{w_{i,j}}(\tilde{w}, \tilde{u})\right) \right|$$
$$\leq \sum_{k=1}^{K} \frac{\left|\langle (\nabla_{w_k} F_{w_{i,j}}(\overline{w}, \overline{u})) \circ \overline{w}_k, v_k - y_k \rangle\right|}{F_{w_{i,j}}(\overline{w}, \overline{u})} + \frac{\left|\langle (\nabla_u F_{w_{i,j}}(\overline{w}, \overline{u})) \circ \overline{u}, x - z \rangle\right|}{F_{w_{i,j}}(\overline{w}, \overline{u})}$$
$$\leq \sum_{k=1}^{K} \frac{\|(\nabla_{w_k} F_{w_{i,j}}(\overline{w}, \overline{u})) \circ \overline{w}_k\|_1}{F_{w_{i,j}}(\overline{w}, \overline{u})} \|v_k - y_k\|_\infty + \frac{\|(\nabla_u F_{w_{i,j}}(\overline{w}, \overline{u})) \circ \overline{u}\|_1}{F_{w_{i,j}}(\overline{w}, \overline{u})} \|x - z\|_\infty$$
$$= \sum_{k=1}^{K} \frac{\langle |\nabla_{w_k} F_{w_{i,j}}(\overline{w}, \overline{u})|, \overline{w}_k \rangle}{F_{w_{i,j}}(\overline{w}, \overline{u})} \|\ln(w_k) - \ln(\tilde{w}_k)\|_\infty + \frac{\langle |\nabla_u F_{w_{i,j}}(\overline{w}, \overline{u})|, \overline{u} \rangle}{F_{w_{i,j}}(\overline{w}, \overline{u})} \|\ln(u) - \ln(\tilde{u})\|_\infty$$
$$\leq \sum_{k=1}^{K} A_{i,k} \|\ln(w_k) - \ln(\tilde{w}_k)\|_\infty + A_{i,K+1} \|\ln(u) - \ln(\tilde{u})\|_\infty.$$

In particular, taking the maximum over $j \in [n_1]$ shows that for every $i \in [K]$ we have

$$\|\ln\left(F_{w_i}(w, u)\right) - \ln\left(F_{w_i}(\tilde{w}, \tilde{u})\right)\|_\infty \leq \sum_{s=1}^{K} A_{i,s} \|\ln(w_s) - \ln(\tilde{w}_s)\|_\infty + A_{i,K+1} \|\ln(u) - \ln(\tilde{u})\|_\infty.$$

A similar argument shows that $\|\ln\left(F_u(w, u)\right) - \ln\left(F_u(\tilde{w}, \tilde{u})\right)\|_\infty$ is upper bounded by

$$\sum_{s=1}^{K} A_{K+1,s} \|\ln(w_s) - \ln(\tilde{w}_s)\|_\infty + A_{K+1,K+1} \|\ln(u) - \ln(\tilde{u})\|_\infty.$$

So, we finally get

$$\mu\left(F(w, u), F(\tilde{w}, \tilde{u})\right) = \sum_{i=1}^{K} \gamma_i \|\ln\left(F_{w_i}(w, u)\right) - \ln\left(F_{w_i}(\tilde{w}, \tilde{u})\right)\|_\infty$$
$$+ \gamma_{K+1} \|\ln\left(F_u(w, u)\right) - \ln\left(F_u(\tilde{w}, \tilde{u})\right)\|_\infty$$
$$\leq \sum_{s=1}^{K} (A^T \gamma)_s \|\ln(w_s) - \ln(\tilde{w}_s)\|_\infty + (A^T \gamma)_{K+1} \|\ln(u) - \ln(\tilde{u})\|_\infty$$
$$\leq U \mu\left((w, u), (\tilde{w}, \tilde{u})\right). \qquad \square$$

Note that, from the Collatz-Wielandt ratio for nonnegative matrices, we know that the constant $U$ in Lemma 4 is lower bounded by the spectral radius $\rho(A)$ of $A$. Indeed, by Theorem 8.1.31 in [10], we know that if $A^T$ has a positive eigenvector $\gamma \in \mathbb{R}_{++}^{K+1}$, then

$$\max_{i \in [K+1]} \frac{(A^T \gamma)_i}{\gamma_i} = \rho(A) = \min_{\tilde{\gamma} \in \mathbb{R}_{++}^{K+1}} \max_{i \in [K+1]} \frac{(A^T \tilde{\gamma})_i}{\tilde{\gamma}_i}. \tag{8}$$

Therefore, in order to obtain the minimal Lipschitz constant $U$ in Lemma 4, we choose the weights of the metric $\mu$ to be the components of $\gamma$. A combination of Theorem 2, Lemma 4 and this observation implies the following result.

**Theorem 3.** *Let* $\Phi \in C^1(V, \mathbb{R}) \cap C^2(B_{++}, \mathbb{R})$ *with* $\nabla \Phi(S_+) \subset V_{++}$. *Let* $G^\Phi \colon B_{++} \to B_{++}$ *be defined as in* (3). *Suppose that there exists a matrix* $A \in \mathbb{R}_+^{(K+1) \times (K+1)}$ *such that* $G^\Phi$ *and* $A$ *satisfies the assumptions of Lemma 4 and* $A^T$ *has a positive eigenvector* $\gamma \in \mathbb{R}_{++}^{K+1}$. *If*

$\rho(A) < 1$, then $\Phi$ has a unique critical point $(w^*, u^*)$ in $S_{++}$ which is the global maximum of the optimization problem (2). Moreover, the sequence $\big((w^k, u^k)\big)_k$ defined for any $(w^0, u^0) \in S_{++}$ as $(w^{k+1}, u^{k+1}) = G^\Phi(w^k, u^k)$, $k \in \mathbb{N}$, satisfies $\lim_{k \to \infty}(w^k, u^k) = (w^*, u^*)$ and

$$\|(w^k, u^k) - (w^*, u^*)\|_\infty \le \rho(A)^k \left( \frac{\mu\big((w^1, u^1), (w^0, u^0)\big)}{\big(1 - \rho(A)\big) \min\big\{ \frac{\gamma_{K+1}}{\rho_u}, \min_{t \in [K]} \frac{\gamma_t}{\rho_w} \big\}} \right) \qquad \forall k \in \mathbb{N},$$

where the weights in the definition of $\mu$ are the entries of $\gamma$.

*Proof.* As $\gamma$ is a positive eigenvector of $A^T$, by (8), we know that $A^T \gamma = \rho(A)\gamma$. It follows from Lemma 4 that $\mu(G^\Phi(w, u), G^\Phi(\tilde{w}, \tilde{u})) < \rho(A)\,\mu\big((w, u), (\tilde{w}, \tilde{u})\big)$ for every $(w, u), (\tilde{w}, \tilde{u}) \in B_{++}$, i.e. $G^\Phi$ is a strict contraction on the complete metric space $(B_{++}, \mu)$ and by the Banach fixed point theorem 2 we know that $G^\Phi$ has a unique fixed point $(w^*, u^*)$ in $S_{++}$. From Lemma 1 we know that $\Phi$ attains its global maximum in $S_{++}$ and, by Lemma 2, this maximum is a fixed point of $G^\Phi$. Hence, $(w^*, u^*)$ is the unique global maximum of $\Phi$ in $S_{++}$. Finally, Theorem 2 implies that

$$\mu\big((w^k, u^k), (w^*, u^*)\big) \le \frac{\rho(A)^k}{1 - \rho(A)} \mu\big((w^1, u^1), (w^0, u^0)\big) \qquad \forall k \in \mathbb{N}.$$

The mean value theorem implies that for every $r \in \mathbb{R}$, we have

$$|e^s - e^t| \le |s - t| \max_{\xi \in (-\infty, r]} e^\xi = e^r |s - t| \qquad \forall s, t \in (-\infty, r].$$

In particular, we have

$$\ln(w_{a,b}^k), \ln(w_{a,b}^*) \in (-\infty, \ln(\rho_w)] \qquad \text{and} \qquad \ln(u_{st}^k), \ln(u_{st}^*) \in (-\infty, \ln(\rho_u)].$$

It follows that

$$\mu\big((w^k, u^k), (w^*, u^*)\big) = \sum_{t=1}^K \gamma_t \|\ln(w_t^k) - \ln(w_t^*)\|_\infty + \gamma_{K+1} \|\ln(u^k) - \ln(u^*)\|_\infty$$

$$\ge \sum_{t=1}^K \frac{\gamma_t}{\rho_w} \|w_t^k - w_t^*\|_\infty + \frac{\gamma_{K+1}}{\rho_u} \|u^k - u^*\|_\infty$$

$$\ge \max\Big\{ \max_{t \in [K]} \frac{\gamma_t}{\rho_w} \|w_t^k - w_t^*\|_\infty, \frac{\gamma_{K+1}}{\rho_u} \|u^k - u^*\|_\infty \Big\}$$

$$\ge \min\Big\{ \frac{\gamma_{K+1}}{\rho_u}, \min_{t \in [K]} \frac{\gamma_t}{\rho_w} \Big\} \|(w^k, u^k) - (w^*, u^*)\|_\infty$$

and thus

$$\|(w^k, u^k) - (w^*, u^*)\|_\infty \le \frac{\mu\big((w^k, u^k), (w^*, u^*)\big)}{\min\big\{ \frac{\gamma_{K+1}}{\rho_u}, \min_{t \in [K]} \frac{\gamma_t}{\rho_w} \big\}}$$

$$\le \rho(A)^k \left( \frac{\mu\big((w^1, u^1), (w^0, u^0)\big)}{\big(1 - \rho(A)\big) \min\big\{ \frac{\gamma_{K+1}}{\rho_u}, \min_{t \in [K]} \frac{\gamma_t}{\rho_w} \big\}} \right). \qquad \square$$

## 4  Application to Neural Networks

In the previous sections we have outlined the proof of our main result for a general objective function satisfying certain properties. The purpose of this section is to prove that the properties hold for our optimization problem for neural networks.

We recall our objective function from (2)

$$\Phi(w, u) = \frac{1}{n} \sum_{i=1}^n \Big[ -L\big(y_i, f(w, u)(x^i)\big) + \sum_{r=1}^K f_r(w, u)(x^i) \Big] + \epsilon\Big( \sum_{r=1}^K \sum_{l=1}^{n_1} w_{r,l} + \sum_{l=1}^{n_1} \sum_{m=1}^d u_{lm} \Big)$$

and the function class we are considering from (1)

$$f_r(x) = f_r(w,u)(x) = \sum_{l=1}^{n_1} w_{r,l} \Big( \sum_{m=1}^{d} u_{lm} x_m \Big)^{\alpha_l},$$

The arbitrarily small $\epsilon$ in the objective is needed to make the gradient strictly positive on the boundary of $V_+$. We note that the assumption $\alpha_i \geq 1$ for every $i \in [n_1]$ is crucial in the following lemma in order to guarantee that $\nabla\Phi$ is well defined on $S_+$.

**Lemma 5.** *Let $\Phi$ be defined as in* (2), *then $\nabla\Phi(w,u)$ is strictly positive for any $(w,u) \in S_+$.*

*Proof.* One can compute

$$\frac{\partial\Phi(w,u)}{\partial w_{a,b}} = \frac{1}{n}\sum_{i=1}^{n}\Big(\delta_{y_i a} - \frac{e^{f_a(x^i)}}{\sum_{j=1}^{K} e^{f_j(x^i)}} + 1\Big)\Big(\sum_{m=1}^{d} u_{bm}x_m^i\Big)^{\alpha_b} + \epsilon,$$

where $\delta_{ij} = \begin{cases} 1 & \text{if } i = j, \\ 0 & \text{else} \end{cases}$ denotes the Kronecker delta, and

$$\frac{\partial\Phi(w,u)}{\partial u_{ab}} = \frac{1}{n}\sum_{i=1}^{n}\sum_{r=1}^{K}\Big(\delta_{y_i r} - \frac{e^{f_r(x^i)}}{\sum_{j=1}^{K} e^{f_j(x^i)}} + 1\Big)\Big(w_{r,a}\alpha_a\big(\sum_{m=1}^{d} u_{am}x_m^i\big)^{\alpha_a - 1} x_b^i\Big) + \epsilon.$$

Note that

$$\delta_{y_i a} - \frac{e^{f_a(x_i)}}{\sum_{j=1}^{K} e^{f_j(x)}} + 1 > 0,$$

and thus using $(w,u) \in S_+$, we deduce that even without the additional $\epsilon$, the gradient would be strictly positive on $S_{++}$ assuming that there exists $k \in [n]$ such that $x^k \neq 0$. However, at the boundary it can happen that the gradient is zero and thus an arbitrarily small $\epsilon$ is sufficent to guarantee that the gradient is strictly positive on $S_{++}$. $\qquad\square$

Next, we derive the matrix $A \in \mathbb{R}^{(K+1)\times(K+1)}$ in order to apply Theorem 3 to $G^\Phi$ with $\Phi$ defined in (2). As discussed in its proof, the matrix $A$ given in the following theorem has a smaller spectral radius than that of Theorem 1. To express this matrix, we consider $\Psi_{p,q}^\alpha : \mathbb{R}_{++}^{n_1} \times \mathbb{R}_{++} \to \mathbb{R}_{++}$ defined for $p,q \in (1,\infty)$ and $\alpha \in \mathbb{R}_{++}^{n_1}$ as

$$\Psi_{p,q}^\alpha(\delta,t) = \left( \Big[ \sum_{l \in J}(\delta_l\, t^{\alpha_l})^{\frac{pq}{q-\overline{\alpha}p}} \Big]^{1-\frac{\overline{\alpha}p}{q}} + \max_{j \in J^c}(\delta_j\, t^{\alpha_j})^p \right)^{1/p}, \tag{9}$$

where $J = \{l \in [n_1] \mid \alpha_l p \leq q\}$, $J^c = \{l \in [n_1] \mid \alpha_l p > q\}$ and $\overline{\alpha} = \min_{l \in J}\alpha_l$.

**Theorem 4.** *Let $\Phi$ be defined as above and $G^\Phi$ be as in* (3). *Set $C_w = \rho_w\, \Psi_{p'_w,p_u}^\alpha(\mathbf{1}, \rho_u\rho_x)$, $C_u = \rho_w\, \Psi_{p'_w,p_u}^\alpha(\alpha, \rho_u\rho_x)$ and $\rho_x = \max_{i\in[n]}\|x^i\|_{p'_u}$. Then $A$ and $G^\Phi$ satisfy all assumptions of Lemma 4 with*

$$A = 2\,\mathrm{diag}\big(p'_w - 1,\ldots,p'_w - 1, p'_u - 1\big)\begin{pmatrix} Q_{w,w} & Q_{w,u} \\ Q_{u,w} & Q_{u,u} \end{pmatrix}$$

*where $Q_{w,w} \in \mathbb{R}_{++}^{K\times K}, Q_{w,u} \in \mathbb{R}_{++}^{K\times 1}, Q_{u,w} \in \mathbb{R}_{++}^{1\times K}$ and $Q_{u,u} \in \mathbb{R}_{++}$ are defined as*

$$\begin{aligned} Q_{w,w} &= 2C_w\mathbf{1}\mathbf{1}^T, & Q_{w,u} &= (2C_u + \|\alpha\|_\infty)\mathbf{1}, \\ Q_{u,w} &= (2C_w + 1)\mathbf{1}^T, & Q_{u,u} &= (2C_u + \|\alpha\|_\infty - 1). \end{aligned}$$

In the supplementary material, we prove that $\Psi_{p,q}^\alpha(\delta,t) \leq \sum_{l=1}^{n_1}\delta_l t^{\alpha_l}$ which yields the weaker bounds $\xi_1, \xi_2$ given in Theorem 1. In particular, this observation combined with Theorems 3 and 4 implies Theorem 1.

*Proof.* We omit in the following the summation intervals as this is clear from the definition of the variables e.g. if $w \in \mathbb{R}^{K\times n_1}$ then $\sum_{a,b} w_{a,b} = \sum_{a=1}^{K}\sum_{b=1}^{n_1} w_{a,b}$. We split the proof in three steps so that we can reuse some of them in the proof of the similar theorem for two hidden layers.

**Proposition 1.** *Suppose there exist $M_{i,j}, C_i > 0$, $i,j \in \{w,u\}$ such that for every $x \in \{x^1, \ldots, x^n\}$ and $r, s \in [K]$, we have*

$$\sum_t w_{s,t} \frac{\partial f_r(x)}{\partial w_{s,t}} \le C_w, \qquad \sum_{a,b} u_{ab} \frac{\partial f_r(x)}{\partial u_{ab}} \le C_u,$$

$$\sum_t w_{s,t} \frac{\partial^2 f_r(x)}{\partial w_{s,t} \partial w_{a,b}} \le M_{w,w} \frac{\partial f_r(x)}{\partial w_{a,b}}, \qquad \sum_{a,b} u_{ab} \frac{\partial^2 f_r(x)}{\partial u_{ab} \partial w_{s,t}} \le M_{w,u} \frac{\partial f(x)}{\partial w_{s,t}},$$

$$\sum_t w_{s,t} \frac{\partial^2 f_r(x)}{\partial w_{s,t} \partial u_{ab}} \le M_{u,w} \frac{\partial f_r(x)}{\partial u_{ab}}, \qquad \sum_{a,b} u_{ab} \frac{\partial^2 f_r(x)}{\partial u_{ab} \partial u_{st}} \le M_{u,u} \frac{\partial f(x)}{\partial u_{st}}.$$

*Then $A$ and $F = G^\Phi$ satisfy all assumptions of Lemma 4 for*

$$A = 2 \operatorname{diag}\left(p'_w - 1, \ldots, p'_w - 1, p'_u - 1\right) \begin{pmatrix} Q_{w,w} & Q_{w,u} \\ Q_{u,w} & Q_{u,u} \end{pmatrix}$$

*where $Q_{w,w} \in \mathbb{R}_{++}^{K \times K}, Q_{w,u} \in \mathbb{R}_{++}^{K \times 1}, Q_{u,w} \in \mathbb{R}_{++}^{1 \times K}, Q_{u,u} \in \mathbb{R}_{++}$ are constant matrices given by*

$$\begin{aligned} Q_{w,w} &= (2C_w + M_{w,w})\mathbf{1}\mathbf{1}^T, & Q_{w,u} &= (2C_u + M_{w,u})\mathbf{1}, \\ Q_{u,w} &= (2C_w + M_{u,w})\mathbf{1}^T, & Q_{u,u} &= (2C_u + M_{u,u}). \end{aligned}$$

*Proof.* In order to save space we make the proof w.r.t. to abstract variables $g, h$, that is $g, h \in \{w_1, \ldots, w_K, u\}$. First of all, note that we have

$$\begin{aligned}
\frac{\partial G^\Phi_{h_a}}{\partial g_s} &= \frac{\partial}{\partial g_s} \frac{\rho_h \left(\frac{\partial \Phi}{\partial h_a}\right)^{p'_h - 1}}{\|\psi_{p'_h}(\nabla_h \Phi)\|_{p_h}} = \frac{\partial}{\partial g_s} \frac{\rho_h \left(\frac{\partial \Phi}{\partial h_a}\right)^{p'_h - 1}}{\|\nabla_h \Phi\|_{p'_h}^{p'_h - 1}} \\
&= (p'_h - 1)\rho_h \left[ \frac{\left(\frac{\partial \Phi}{\partial h_a}\right)^{p'_h - 2} \frac{\partial^2 \Phi}{\partial g_s \partial h_a}}{\|\nabla_h \Phi\|_{p'_h}^{p'_h - 1}} - \frac{\left(\frac{\partial \Phi}{\partial h_a}\right)^{p'_h - 1} \|\nabla_h \Phi\|_{p'_h}^{-1} \sum_c \left(\frac{\partial \Phi}{\partial h_c}\right)^{p'_h - 1} \frac{\partial^2 \Phi}{\partial g_s \partial h_c}}{\|\nabla_h \Phi\|_{p'_h}^{2p'_h - 2}} \right] \\
&= \left[ \frac{\frac{\partial^2 \Phi}{\partial g_s \partial h_a}}{\frac{\partial \Phi}{\partial h_a}} - \frac{\sum_c \left(\frac{\partial \Phi}{\partial h_c}\right)^{p'_h - 1} \frac{\partial^2 \Phi}{\partial g_s \partial h_c}}{\|\nabla_h \Phi\|_{p'_h}^{p'_h}} \right] (p'_h - 1) G^\Phi_{h_a},
\end{aligned}$$

where $\rho_h = \rho_w$ if $h \in \{w_1, \ldots, w_K\}$ and $\rho = \rho_u$ if $h = u$. Thus,

$$\sum_s g_s \left| \frac{\partial G^\Phi_{h_a}}{\partial g_s} \right| = \left[ \sum_s g_s \left| \frac{\frac{\partial^2 \Phi}{\partial g_s \partial h_a}}{\frac{\partial \Phi}{\partial h_a}} - \frac{\sum_c \left(\frac{\partial \Phi}{\partial h_c}\right)^{p'_h - 1} \frac{\partial^2 \Phi}{\partial g_s \partial h_c}}{\|\nabla_h \Phi\|_{p'_h}^{p'_h}} \right| \right] (p'_h - 1) G^\Phi_{h_a}.$$

Now, suppose that there exists $R_{h,g} > 0$ such that

$$\sum_s g_s \left| \frac{\partial^2 \Phi}{\partial g_s \partial h_a} \right| \le R_{h,g} \frac{\partial \Phi}{\partial h_a}, \tag{10}$$

then we get

$$A_{h,g} = (p'_h - 1) \max_a \sum_s g_s \left| \frac{\frac{\partial^2 \Phi}{\partial g_s \partial h_a}}{\frac{\partial \Phi}{\partial h_a}} - \frac{\sum_c \left(\frac{\partial \Phi}{\partial h_c}\right)^{p'_h - 1} \frac{\partial^2 \Phi}{\partial g_s \partial h_c}}{\|\nabla_h \Phi\|_{p'_h}^{p'_h}} \right|$$

$$\leq (p' - 1) \max_a \sum_s g_s \left| \frac{\frac{\partial^2 \Phi}{\partial g_s \partial h_a}}{\frac{\partial \Phi}{\partial h_a}} \right| + (p'_h - 1) \sum_s g_s \left| \frac{\sum_c \left(\frac{\partial \Phi}{\partial h_c}\right)^{p'_h - 1} \frac{\partial^2 \Phi}{\partial g_s \partial h_c}}{\|\nabla_h \Phi\|_{p'_h}^{p'_h}} \right|$$

$$\leq (p'_h - 1) \max_a \sum_s g_s \left| \frac{\frac{\partial^2 \Phi}{\partial g_s \partial h_a}}{\frac{\partial \Phi}{\partial h_a}} \right| + (p'_h - 1) \frac{\sum_c \left(\frac{\partial \Phi}{\partial h_c}\right)^{p'_h - 1} \sum_s g_s \left| \frac{\partial^2 \Phi}{\partial g_s \partial h_c} \right|}{\|\nabla_h \Phi\|_{p'_h}^{p'_h}}$$

$$\leq (p'_h - 1) R_{h,g} + (p'_h - 1) R_{h,g} \frac{\sum_c \left(\frac{\partial \Phi}{\partial h_c}\right)^{p'_h}}{\|\nabla_h \Phi\|_{p'_h}^{p'_h}}$$

$$= 2(p'_h - 1) R_{h,g}.$$

It follows that, if we define $A_{h,g} = 2(p'_h - 1) R_{h,g}$ for all $g, h \in \{w_1, \ldots, w_K, u\}$, then $A$ and $G^\Phi$ satisfy all assumptions of Lemma 4. In order to conclude the proof, we show that $R_{h,g} = 2C_g + M_{h,g}$. We compute the derivatives of the cross-entropy loss

$$-\frac{\partial L}{\partial f_r(x)} = \delta_{yr} - \frac{e^{f_r(x)}}{\sum_{j=1}^K e^{f_j(x)}},$$

$$-\frac{\partial^2 L}{\partial f_q(x) \partial f_r(x)} = -\delta_{qr} \frac{e^{f_r(x)}}{\sum_{j=1}^K e^{f_j(x)}} + \frac{e^{f_r(x)} e^{f_q(x)}}{\left(\sum_{j=1}^K e^{f_j(x)}\right)^2},$$

We get for the derivatives of the objective with respect to the abstract variables $g, h$

$$\frac{\partial \Phi}{\partial g_s} = \frac{1}{n} \sum_{i=1}^n \sum_{r=1}^K \left( -\frac{\partial L}{\partial f_r} \Big|_{f(x^i)} + 1 \right) \frac{\partial f_r}{\partial g_s} \Big|_{x^i} + \epsilon \tag{11}$$

and

$$\frac{\partial \Phi}{\partial g_s \partial h_a} = \frac{1}{n} \sum_{i=1}^n \sum_{q,r=1}^K \left( -\frac{\partial^2 L}{\partial f_q \partial f_r} \Big|_{f(x^i)} \right) \frac{\partial f_r}{\partial h_a} \Big|_{x^i} \frac{\partial f_q}{\partial g_s} \Big|_{x^i} + \frac{1}{n} \sum_{i=1}^n \sum_{r=1}^K \left( -\frac{\partial L}{\partial f_r} \Big|_{f(x^i)} + 1 \right) \frac{\partial^2 f_r}{\partial g_s \partial h_a} \Big|_{x^i}$$

We then can upper bound

$$\sum_s g_s \left| \sum_{q,r=1}^K \left( -\frac{\partial^2 L}{\partial f_q \partial f_r} \Big|_{f(x^i)} \right) \frac{\partial f_r}{\partial h_a} \Big|_{x^i} \frac{\partial f_q}{\partial g_s} \Big|_{x^i} \right|$$

$$\leq \sum_{q,r=1}^K \left| \left( \frac{\partial^2 L}{\partial f_q \partial f_r} \Big|_{f(x^i)} \right) \right| \frac{\partial f_r}{\partial h_a} \Big|_{x^i} \sum_s g_s \frac{\partial f_q}{\partial g_s} \Big|_{x^i}$$

$$\leq \max_{l=1,\ldots,K} \sum_s g_s \frac{\partial f_l}{\partial g_s} \Big|_{x^i} \sum_{r=1}^K \frac{e^{f_r(x^i)}}{\sum_{j=1}^K e^{f_j(x^i)}} \left[ \sum_{q \neq r} \frac{e^{f_q(x^i)}}{\sum_{j=1}^K e^{f_j(x)}} + \left( 1 - \frac{e^{f_r(x^i)}}{\sum_{j=1}^K e^{f_j(x^i)}} \right) \right] \frac{\partial f_r}{\partial h_a} \Big|_{x^i}$$

$$= 2 \max_{l=1,\ldots,K} \sum_s g_s \frac{\partial f_l}{\partial g_s} \Big|_{x^i} \sum_{r=1}^K \frac{e^{f_r(x^i)}}{\sum_{j=1}^K e^{f_j(x^i)}} \left( 1 - \frac{e^{f_r(x^i)}}{\sum_{j=1}^K e^{f_j(x^i)}} \right) \frac{\partial f_r}{\partial h_a} \Big|_{x^i}$$

$$\leq 2 \max_{l=1,\ldots,K} \sum_s g_s \frac{\partial f_l}{\partial g_s} \Big|_{x^i} \sum_{r=1}^K \left( 1 - \frac{e^{f_r(x^i)}}{\sum_{j=1}^K e^{f_j(x^i)}} + \delta_{y_i r} \right) \frac{\partial f_r}{\partial h_a} \Big|_{x^i}$$

where we have used the non-negativity of the derivatives of the function class in the first step and

$$\sum_{q\neq r}\frac{e^{f_q(x)}}{\sum_{j=1}^K e^{f_j(x)}}=1-\frac{e^{f_r(x)}}{\sum_{j=1}^K e^{f_j(x)}}.$$

Now, note that

$$\sum_s g_s\frac{1}{n}\sum_{i=1}^n\sum_{r=1}^K\Big(-\frac{\partial L}{\partial f_r}\Big|_{f(x^i)}+1\Big)\frac{\partial^2 f_r}{\partial h_a\partial g_s}\Big|_{x^i}\leq M_{h,g}\frac{1}{n}\sum_{i=1}^n\sum_{r=1}^K\Big(-\frac{\partial L}{\partial f_r}\Big|_{f(x^i)}+1\Big)\frac{\partial f_r}{\partial h_a}\Big|_{x^i}$$

$$\leq M_{h,g}\frac{\partial\Phi}{\partial h_a}.$$

This shows that

$$\sum_s g_s\Big|\frac{\partial^2\Phi}{\partial g_s\partial h_a}\Big|$$

$$\leq 2\max_{k,l}\sum_s g_s\frac{\partial f_l}{\partial g_s}\Big|_{x^k}\frac{1}{n}\sum_{i=1}^n\sum_{r=1}^K\Big(1-\frac{e^{f_r(x^i)}}{\sum_{j=1}^K e^{f_j(x^i)}}+\delta_{y_i r}\Big)\frac{\partial f_r}{\partial h_a}\Big|_{x^i}+M_{h,g}\frac{\partial\Phi}{\partial h_a}$$

$$\leq\Big[2\max_{k,l}\sum_s g_s\frac{\partial f_l}{\partial g_s}\Big|_{x^k}+M_{h,g}\Big]\frac{\partial\Phi}{\partial h_a}\leq(2C_g+M_{h,g})\frac{\partial\Phi}{\partial h_a}.$$

Thus, we get $R_{h,g}\leq 2C_g+M_{h,g}$. $\qquad\square$

The following lemma is useful for the estimation of the bounds $C_w,C_u$ in Proposition 1.

**Lemma 6.** *Let $\alpha\in\mathbb{R}_{++}^r$, $p,p_u\in[1,\infty)$ and $\Psi_{p,p_u}^\alpha:\mathbb{R}_{++}^r\times\mathbb{R}_{++}\to\mathbb{R}_{++}$ defined as in (9). Then, for $x\in\mathbb{R}_+^s$ and $u\in\mathbb{R}_{++}^{r\times s}$ satisfying $\|u\|_{p_u}\leq\rho_u$, we have*

$$\Big[\sum_l\delta_l^p\Big(\sum_m u_{lm}x_m\Big)^{p\alpha_l}\Big]^{1/p}\leq\Psi_{p,p_u}^\alpha(\delta,\rho_u\|x\|_{p_u'}).$$

*Moreover, if $(\delta,t)\leq(\tilde\delta,\tilde t)$ then $\Psi_{p,p_u}^\alpha(\delta,t)\leq\Psi_{p,p_u}^\alpha(\tilde\delta,\tilde t)\leq\sum_{l=1}^r\tilde\delta_l\tilde t^{\alpha_l}$.*

*Proof.* Let $J=\{l\in[r]\mid\alpha_l p<p_u\}$ and $J^c=\{l\in[r]\mid\alpha_l p\geq p_u\}$, we start with some observations. On the one hand, as $\|u\|_{p_u}\leq\rho_u$, we have

$$\sum_{l\in J^c}\Big(\frac{\sum_m u_{lm}^{p_u}}{\rho_u^{p_u}}\Big)^{p\alpha_l/p_u}\leq\sum_{l\in J^c}\frac{\sum_m u_{lm}^{p_u}}{\rho_u^{p_u}}\leq\frac{\|u\|_{p_u}^{p_u}}{\rho_u^{p_u}}\leq 1.$$

On the other hand, if $\overline\alpha=\min_{l\in J}\alpha_l$, then $\overline p=\frac{p_u}{\overline\alpha p}>1$, $\overline p'=\frac{p_u}{p_u-\overline\alpha p}$, and

$$\sum_{l\in J}\Big(\frac{\sum_m u_{lm}^{p_u}}{\rho_u^{p_u}}\Big)^{\overline p(p\alpha_l/p_u)}=\sum_{l\in J}\Big(\frac{\sum_m u_{lm}^{p_u}}{\rho_u^{p_u}}\Big)^{\frac{\alpha_l}{\overline\alpha}}\leq\sum_{l\in J}\Big(\frac{\sum_m u_{lm}^{p_u}}{\rho_u^{p_u}}\Big)\leq 1.$$

It follows that

$$\sum_l\delta_l^p\Big(\sum_m u_{lm}x_m\Big)^{p\alpha_l}\leq\sum_l\Big(\sum_m u_{lm}^{p_u}\Big)^{\frac{p\alpha_l}{p_u}}\delta_l^p\|x\|_{p_u'}^{p\alpha_l}$$

$$=\sum_{l\in J}\Big(\frac{\sum_m u_{lm}^{p_u}}{\rho_u^{p_u}}\Big)^{p\alpha_l/p_u}\delta_l^p(\rho_u\|x\|_{p_u'})^{p\alpha_l}+\sum_{l\in J^c}\Big(\frac{\sum_m u_{lm}^{p_u}}{\rho_u^{p_u}}\Big)^{p\alpha_l/p_u}\delta_l^p(\rho_u\|x\|_{p_u'})^{p\alpha_l}$$

$$\leq\sum_{l\in J}\Big(\frac{\sum_m u_{lm}^{p_u}}{\rho_u^{p_u}}\Big)^{p\alpha_l/p_u}\delta_l^p(\rho_u\|x\|_{p_u'})^{p\alpha_l}+\max_{j\in J^c}\delta_j^p(\rho_u\|x\|_{p_u'})^{p\alpha_j}\sum_{l\in J^c}\Big(\frac{\sum_m u_{lm}^{p_u}}{\rho_u^{p_u}}\Big)^{p\alpha_l/p_u}$$

$$\leq\Big[\sum_{l\in J}\Big(\frac{\sum_m u_{lm}^{p_u}}{\rho_u^{p_u}}\Big)^{\overline p(p\alpha_l/p_u)}\Big]^{1/\overline p}\Big[\sum_{l\in J}(\delta_l^p(\rho_u\|x\|_{p_u'})^{p\alpha_l})^{\overline p'}\Big]^{1/\overline p'}+\max_{j\in J^c}\delta_j^p(\rho_u\|x\|_{p_u'})^{p\alpha_j}$$

$$\leq\Big[\sum_{l\in J}(\delta_l^p(\rho_u\|x\|_{p_u'})^{p\alpha_l})^{\overline p'}\Big]^{1/\overline p'}+\max_{j\in J^c}\delta_j^p(\rho_u\|x\|_{p_u'})^{p\alpha_j}=\big[\Psi_{p,p_u}^\alpha(\delta,\rho_u\|x\|_{p_u'})\big]^p$$

In particular, we see that

$$\Psi_{p,p_u}^{\alpha}(\delta, t) = \left( \left[ \sum_{l \in J} (\delta_l\, t^{\alpha_l})^{p\,\overline{p}'} \right]^{1/\overline{p}'} + \max_{j \in J^c} (\delta_j\, t^{\alpha_j})^p \right)^{1/p}.$$

Finally, let $(\delta, t) \leq (\tilde{\delta}, \tilde{t})$. By monotonicity of $\|\cdot\|_{\overline{p}}$ and $\|\cdot\|_p$, we have $\Psi_{p,p_u}^{\alpha}(\delta, t) \leq \Psi_{p,p_u}^{\alpha}(\tilde{\delta}, \tilde{t})$. Now, using $\|\cdot\|_p \leq \|\cdot\|_1$ and $\|\cdot\|_{p\,\overline{p}'} \leq \|\cdot\|_1$, we get

$$\begin{aligned}
\Psi_{p,p_u}^{\alpha}(\delta, t) &= \left( \left( \left[ \sum_{l \in J} (\delta_l\, t^{\alpha_l})^{p\,\overline{p}'} \right]^{1/(p\,\overline{p}')} \right)^p + \max_{j \in J^c} (\delta_j\, t^{\alpha_j})^p \right)^{1/p} \\
&\leq \left[ \sum_{l \in J} (\delta_l\, t^{\alpha_l})^{p\,\overline{p}'} \right]^{1/(p\,\overline{p}')} + \max_{j \in J^c} \delta_j\, t^{\alpha_j} \leq \sum_{l \in J} \delta_l\, t^{\alpha_l} + \max_{j \in J^c} \delta_j\, t^{\alpha_j} \\
&\leq \sum_{l \in [r]} \delta_l\, t^{\alpha_l}. \qquad\qquad\qquad\qquad\qquad\qquad\qquad\qquad\qquad\quad \square
\end{aligned}$$

Finally, using the Lemma above, we explicit the bounds of Proposition 1.

**Lemma 7.** *Let $f$ be defined as in (1) and let $\rho_x = \max_{i=1 \in [n]} \|x^i\|_{p'_u}$, then the bounds in Proposition 1 are given by $C_w = \rho_w \Psi_{p'_w,p_u}^{\alpha}(\mathbf{1}, \rho_u \rho_x)$, $C_u = \rho_w \Psi_{p'_w,p_u}^{\alpha}(\alpha, \rho_u \rho_x)$, $M_{w,w} = 0$, $M_{w,u} = \|\alpha\|_{\infty}$, $M_{u,w} = 1$ and $M_{u,u} = \|\alpha\|_{\infty} - 1$.*

*Proof.* Let $x \in \{x^1, \ldots, x^n\}$ and, for $(w, u) \in B_{++}$, we have

$$\begin{aligned}
\frac{\partial f_r(x)}{\partial w_{a,b}} &= \delta_{ra} \Big( \sum_m u_{bm} x_m \Big)^{\alpha_b}, \\
\frac{\partial f_r(x)}{\partial u_{ab}} &= w_{r,a} \alpha_a \Big( \sum_m u_{am} x_m \Big)^{\alpha_a - 1} x_b, \\
\frac{\partial^2 f_r(x)}{\partial w_{s,t} \partial w_{a,b}} &= 0, \\
\frac{\partial^2 f_r(x)}{\partial w_{s,t} \partial u_{ab}} &= \delta_{rs} \delta_{at} \alpha_a \Big( \sum_m u_{am} x_m \Big)^{\alpha_a - 1} x_b, \\
\frac{\partial^2 f_r(x)}{\partial u_{st} \partial u_{ab}} &= \delta_{as} w_{r,a} \alpha_a (\alpha_a - 1) \Big( \sum_m u_{am} x_m \Big)^{\alpha_a - 2} x_b x_t.
\end{aligned}$$

With Lemma 6, we have

$$\begin{aligned}
\sum_b w_{a,b} \frac{\partial f_r(x)}{\partial w_{a,b}} &= \delta_{ra} \sum_b w_{a,b} \Big( \sum_m u_{bm} x_m \Big)^{\alpha_b} \leq \delta_{ra} \|w_a\|_{p_w} \left( \sum_b \Big( \sum_m u_{bm} x_m \Big)^{\alpha_b p'_w} \right)^{1/p'_w} \\
&\leq \delta_{ra} \|w_a\|_{p_w} \Psi_{p'_w,p_u}^{\alpha}(\mathbf{1}, \rho_u \|x\|_{p'_u}) \leq \rho_w \Psi_{p'_w,p_u}^{\alpha}(\mathbf{1}, \rho_u \rho_x) = C_w,
\end{aligned}$$

and

$$\begin{aligned}
\sum_{a,b} u_{ab} \frac{\partial f_r(x)}{\partial u_{ab}} &\leq \sum_a w_{r,a} \alpha_a \Big( \sum_m u_{am} x_m \Big)^{\alpha_a} \leq \|w_r\|_{p_w} \Psi_{p'_w,p_u}^{\alpha}(\alpha, \rho_u \|x\|_{p'_u}) \\
&\leq \rho_w \Psi_{p'_w,p_u}^{\alpha}(\alpha, \rho_u \rho_x) = C_u.
\end{aligned}$$

Furthermore,

$$\sum_t w_{s,t}\frac{\partial^2 f_r(x)}{\partial w_{s,t}\partial w_{a,b}} = 0 \qquad \implies \quad M_{w,w} = 0$$

$$\sum_{a,b} u_{ab}\frac{\partial^2 f_r(x)}{\partial u_{ab}\partial w_{s,t}} = \delta_{rs}\alpha_t\Big(\sum_m u_{tm}x_m\Big)^{\alpha_t} = \alpha_t\frac{\partial f(x)}{\partial w_{s,t}} \leq \|\alpha\|_\infty\frac{\partial f(x)}{\partial w_{s,t}}$$

$$\implies \quad M_{w,u} = \|\alpha\|_\infty$$

$$\sum_t w_{s,t}\frac{\partial^2 f_r(x)}{\partial w_{s,t}\partial u_{ab}} = \delta_{rs}w_{s,a}\alpha_a\Big(\sum_m u_{am}x_m\Big)^{\alpha_a-1}x_b = \delta_{sr}\frac{\partial f_r(x)}{\partial u_{ab}} \leq \frac{\partial f_r(x)}{\partial u_{ab}}$$

$$\implies \quad M_{u,w} = 1$$

$$\sum_{a,b} u_{ab}\frac{\partial^2 f_r(x)}{\partial u_{st}\partial u_{ab}} = w_{r,s}\alpha_s(\alpha_s-1)\Big(\sum_m u_{sm}x_m\Big)^{\alpha_s-1}x_t = (\alpha_s-1)\frac{\partial f_r(x)}{\partial u_{st}}$$

$$\leq \|\alpha-\mathbf{1}\|_\infty\frac{\partial f_r(x)}{\partial u_{st}} \qquad \implies \quad M_{u,u} = \|\alpha-\mathbf{1}\|_\infty.$$

Finally, as $\alpha_i \geq 1$ for every $i \in [n_1]$, we have $\|\alpha-\mathbf{1}\|_\infty = \|\alpha\|_\infty - 1$. $\qquad\square$

Combining Lemma 7 and Proposition 1 concludes the proof. Finally, we note that the upper bound on $\Psi_{p,q}^\alpha$ proved in Lemma 6 implies that the matrix $A$ in Theorem 4 is componentwise smaller or equal than that of Theorem 1 and thus has a smaller spectral radius by Corollary 3.30 [3]. $\qquad\square$

## 4.1 Neural networks with two hidden layers

We show how to extend our framework for neural networks with 2 hidden layers. In future work we will consider the general case. We briefly explain the major changes. Let $n_1, n_2 \in \mathbb{N}$ and $\alpha \in \mathbb{R}_{++}^{n_1}, \beta \in \mathbb{R}_{++}^{n_2}$ with $\alpha_i, \beta_j \geq 1$ for all $i \in [n_1], j \in [n_2]$, our function class is:

$$f_r(x) = f_r(w,v,u)(x) = \sum_{l=1}^{n_2} w_{r,l}\Big(\sum_{m=1}^{n_1} v_{lm}\Big(\sum_{s=1}^{d} u_{ms}x_s\Big)^{\alpha_m}\Big)^{\beta_l}$$

and the optimization problem becomes

$$\max_{(w,v,u)\in S_+}\Phi(w,v,u) \qquad \text{where} \qquad V_+ = \mathbb{R}_+^{K\times n_2}\times\mathbb{R}_+^{n_2\times n_1}\times\mathbb{R}_+^{n_1\times d}, \qquad (12)$$

$S_+ = \{(w_1,\dots,w_K,v,u)\in V_+ \mid \|w_i\|_{p_w} = \rho_w, \|v\|_{p_v} = \rho_v, \|u\|_{p_u} = \rho_u\}$ and

$$\Phi(w,v,u) = \frac{1}{n}\sum_{i=1}^n\Big[-L\big(y_i, f(x^i)\big)+\sum_{r=1}^K f_r(x^i)\Big]+\epsilon\Big(\sum_{r=1}^K\sum_{l=1}^{n_2} w_{r,l}+\sum_{l=1}^{n_2}\sum_{m=1}^{n_1} v_{lm}+\sum_{m=1}^{n_1}\sum_{s=1}^d u_{ms}\Big).$$

The map $G^\Phi\colon S_{++}\to S_{++} = \{z\in S_+ \mid z > 0\}$, $G^\Phi = (G_{w_1}^\Phi,\dots,G_{w_K}^\Phi, G_v^\Phi, G_u^\Phi)$, becomes

$$G_{w_i}^\Phi(w,v,u) = \rho_w\frac{\psi_{p'_w}(\nabla_{w_i}\Phi(w,u))}{\|\psi_{p'_w}(\nabla_{w_i}\Phi(w,v,u))\|_{p_w}} \qquad \forall i\in[K] \qquad (13)$$

and

$$G_v^\Phi(w,v,u) = \rho_v\frac{\psi_{p'_v}(\nabla_v\Phi(w,v,u))}{\|\psi_{p'_v}(\nabla_v\Phi(w,v,u))\|_{p_v}}, \qquad G_u^\Phi(w,v,u) = \rho_u\frac{\psi_{p'_u}(\nabla_u\Phi(w,v,u))}{\|\psi_{p'_u}(\nabla_u\Phi(w,v,u))\|_{p_u}}.$$

We have the following equivalent of Theorem 1 for 2 hidden layers.

**Theorem 5.** *Let $\{x^i, y_i\}_{i=1}^n \subset \mathbb{R}_+^d \times [K]$, $p_w, p_v, p_u \in (1,\infty)$, $\rho_w, \rho_v, \rho_u > 0$, $n_1, n_2 \in \mathbb{N}$ and $\alpha \in \mathbb{R}_{++}^{n_1}, \beta \in \mathbb{R}_{++}^{n_2}$ with $\alpha_i, \beta_j \geq 1$ for all $i \in [n_1], j \in [n_2]$. Let $\rho_x = \max_{i\in[n]}\|x^i\|_{p'_u}$,*

$$\theta = \rho_v\Psi_{p'_v,p_u}^\alpha(\mathbf{1},\rho_u\rho_x), \quad C_w = \rho_w\Psi_{p'_w,p_v}^\beta(\mathbf{1},\theta), \quad C_v = \rho_w\Psi_{p'_w,p_v}^\beta(\beta,\theta), \quad C_u = \|\alpha\|_\infty C_v,$$

*and define* $A \in \mathbb{R}_{++}^{(K+2)\times(K+2)}$ *as*

$$
\begin{aligned}
A_{m,l} &= 4(p_w' - 1)C_w, & A_{m,K+1} &= 2(p_w' - 1)(2C_v + \|\beta\|_\infty) \\
A_{m,K+2} &= 2(p_w' - 1)\big(2C_u + \|\alpha\|_\infty\|\beta\|_\infty\big), & A_{K+1,l} &= 2(p_v' - 1)\big(2C_w + 1\big) \\
A_{K+1,K+1} &= 2(p_v' - 1)\big(2C_v + \|\beta\|_\infty - 1\big), & A_{K+1,K+2} &= 2(p_v' - 1)\big(2C_u + \|\alpha\|_\infty\|\beta\|_\infty\big) \\
A_{K+2,l} &= 2(p_u' - 1)(2C_w + 1), & A_{K+2,K+1} &= 2(p_u' - 1)(2C_v + \|\beta\|_\infty), \\
A_{K+2,K+2} &= 2(p_u' - 1)(2C_u + \|\alpha\|_\infty\|\beta\|_\infty - 1) & & \forall m,l \in [K].
\end{aligned}
$$

*If $\rho(A) < 1$, then (12) has a unique global maximizer $(w^*, v^*, u^*) \in S_{++}$. Moreover, for every $(w^0, v^0, u^0) \in S_{++}$, there exists $R > 0$ such that*

$$
\lim_{k\to\infty}(w^k, v^k, u^k) = (w^*, v^*, u^*) \qquad and \qquad \|(w^k, v^k, u^k) - (w^*, v^*, u^*)\|_\infty \le R\,\rho(A)^k \quad \forall k \in \mathbb{N}
$$

*where $(w^{k+1}, v^{k+1}, u^{k+1}) = G^\Phi(w^k, v^k, u^k)$ for every $k \in \mathbb{N}$ and $G^\Phi$ is defined as in (13).*

*Proof.* The proof of Theorem 3, can be extended by considering the weighted Thompson metric $\mu\colon V_{++} \times V_{++} \to \mathbb{R}_+$ defined as

$$
\mu\big((w, v, u), (\tilde w, \tilde v, \tilde u)\big) = \sum_{i=1}^K \gamma_i \|\ln(w_i) - \ln(\tilde w_i)\|_\infty + \gamma_{K+1}\|\ln(v) - \ln(\tilde v)\|_\infty + \gamma_{K+2}\|\ln(u) - \ln(\tilde u)\|.
$$

In particular, when $\rho(A) < 1$, the convergence rate becomes

$$
\|(w^k, v^k, u^k) - (w^*, v^*, u^*)\|_\infty \le \rho(A)^k \left( \frac{\mu\big((w^1, v^1, u^1), (w^0, v^0, u^0)\big)}{\big(1 - \rho(A)\big)\min\big\{\frac{\gamma_{K+2}}{\rho_u}, \frac{\gamma_{K+1}}{\rho_v}, \min_{t\in[K]}\frac{\gamma_t}{\rho_w}\big\}} \right) \quad \forall k \in \mathbb{N},
$$

where the weights in the definition of $\mu$ are the components of the positive eigenvector of $A^T$. As $A^T \in \mathbb{R}_{++}^{(K+2)\times(K+2)}$, this vector always exists by the Perron-Frobenius theorem (see for instance Theorem 8.4.4 in [10]). Proposition 1 generalizes straightforwardly. So, we need to compute the corresponding quantities $C_g, M_{g,h} > 0$ for $g, h \in \{w_1, \ldots, w_K, u, v\}$. To shorten notations, we write

$$
(ux)^\alpha = \big((ux)_1^{\alpha_1}, \ldots, (ux)_{n_1}^{\alpha_{n_1}}\big) \qquad and \qquad (v(ux)^\alpha)^\beta = \big((v(ux)^\alpha)_1^{\beta_1}, \ldots, (v(ux)^\alpha)_{n_2}^{\beta_{n_2}}\big),
$$

where

$$
(ux)_i = \sum_{t=1}^d u_{it}x_t \qquad and \qquad (v(ux)^\alpha)_j = \sum_{s=1}^{n_1} v_{js}(ux)_s^{\alpha_s} \qquad \forall i \in [n_1],\, j \in [n_2].
$$

Again, we omit the summation intervals in the following. First, we compute

$$
\frac{\partial f_r(x)}{\partial w_{a,b}} = \delta_{ra}(v(ux)^\alpha)_b^{\beta_b}
$$

$$
\frac{\partial f_r(x)}{\partial v_{ab}} = w_{r,a}\beta_a(v(ux)^\alpha)_a^{\beta_a - 1}(ux)_b^{\alpha_b}
$$

$$
\frac{\partial f_r(x)}{\partial u_{ab}} = \sum_j w_{r,j}\beta_j(v(ux)^\alpha)_j^{\beta_j - 1}v_{ja}\alpha_a(ux)_a^{\alpha_a - 1}x_b
$$

Hence

$$
\sum_b w_{a,b}\frac{\partial f_r(x)}{\partial w_{a,b}} = \delta_{ra}\sum_b w_{r,b}(v(ux)^\alpha)_b^{\beta_b} = \delta_{ra}f_r(x)
$$

$$
\sum_{a,b} v_{ab}\frac{\partial f_r(x)}{\partial v_{ab}} = \sum_a w_{r,a}\beta_a(v(ux)^\alpha)_a^{\beta_a - 1}\sum_b v_{ab}(ux)_b^{\alpha_b} = \sum_a \beta_a w_{r,a}(v(ux)^\alpha)_a^{\beta_a}
$$

$$
\sum_{a,b} u_{ab}\frac{\partial f_r(x)}{\partial u_{ab}} = \sum_j w_{r,j}\beta_j(v(ux)^\alpha)_j^{\beta_j - 1}\sum_a v_{ja}\alpha_a(ux)_a^{\alpha_a - 1}\sum_b u_{ab}x_b
$$

$$
= \sum_j \beta_j w_{r,j}(v(ux)^\alpha)_j^{\beta_j - 1}\sum_a \alpha_a v_{ja}(ux)_a^{\alpha_a}.
$$

It follows with Lemma 6 that, with $\theta = \rho_v \, \Psi^{\alpha}_{p'_v, p_u}(\mathbf{1}, \rho_u \rho_x)$, we have

$$\sum_b w_{a,b} \frac{\partial f_r(x)}{\partial w_{a,b}} = \delta_{ra} \sum_b w_{a,b}(v(ux)^{\alpha})_b^{\beta_b} \leq \|w_a\|_{p_w} \Big( \sum_b (v(ux)^{\alpha})_b^{\beta_b p'_w} \Big)^{1/p'_w}$$

$$\leq \|w_a\|_{p_w} \Psi^{\beta}_{p'_w, p_v}(\mathbf{1}, \rho_v \|(ux)^{\alpha}\|_{p'_v}) \leq \|w_a\|_{p_w} \Psi^{\beta}_{p'_w, p_v}\big(\mathbf{1}, \rho_v \, \Psi^{\alpha}_{p'_v, p_u}(\mathbf{1}, \rho_u \|x\|_{p'_u})\big)$$

$$\leq \rho_w \Psi^{\beta}_{p'_w, p_v}(\mathbf{1}, \theta) = C_w$$

$$\sum_{a,b} v_{ab} \frac{\partial f_r(x)}{\partial v_{ab}} = \sum_a \beta_a w_{r,a}(v(ux)^{\alpha})_a^{\beta_a} \leq \|w_r\|_{p_w} \Big( \sum_a \big[\beta_a (v(ux)^{\alpha})_a^{\beta_a}\big]^{p'_w} \Big)^{1/p'_w}$$

$$\leq \|w_r\|_{p_w} \Psi^{\beta}_{p'_w, p_v}(\beta, \rho_v \|(ux)^{\alpha}\|_{p'_v}) \leq \rho_w \Psi^{\beta}_{p'_w, p_v}(\beta, \theta) = C_v$$

$$\sum_{a,b} u_{ab} \frac{\partial f_r(x)}{\partial u_{ab}} = \sum_j \beta_j w_{r,j}(v(ux)^{\alpha})_j^{\beta_j - 1} \sum_a \alpha_a v_{ja}(ux)_a^{\alpha_a} \leq \|\alpha\|_{\infty} \sum_j \beta_j w_{r,j}(v(ux)^{\alpha})_j^{\beta_j}$$

$$\leq \|\alpha\|_{\infty} C_v = C_u.$$

Now, for the bound involving the second derivatives, we have

$$\frac{\partial^2 f_r(x)}{\partial w_{s,t} \partial w_{a,b}} = 0 \qquad \Longrightarrow \qquad M_{w,w} = 0,$$

and

$$\frac{\partial^2 f_r(x)}{\partial v_{st} \partial v_{ab}} = \delta_{sa} w_{r,a} \beta_a (\beta_a - 1)(v(ux)^{\alpha})_a^{\beta_a - 2}(ux)_b^{\alpha_b}(ux)_t^{\alpha_t}$$

so that

$$\sum_{s,t} v_{st} \frac{\partial^2 f_r(x)}{\partial v_{st} \partial v_{ab}} = w_{r,a} \beta_a (\beta_a - 1)(v(ux)^{\alpha})_a^{\beta_a - 2}(ux)_b^{\alpha_b} \sum_t v_{at}(ux)_t^{\alpha_t}$$

$$= w_{r,a}(\beta_a - 1)\beta_a (v(ux)^{\alpha})_a^{\beta_a - 1}(ux)_b^{\alpha_b}$$

$$= (\beta_a - 1)\frac{\partial f_r(x)}{\partial v_{ab}} \leq \|\beta - \mathbf{1}\|_{\infty} \frac{\partial f_r(x)}{\partial v_{ab}} \qquad \Longrightarrow \qquad M_{v,v} = \|\beta - \mathbf{1}\|_{\infty}.$$

Moreover, we have

$$\frac{\partial^2 f_r(x)}{\partial u_{st} \partial u_{ab}} = \sum_j w_{r,j} \beta_j v_{ja} \alpha_a (ux)_a^{\alpha_a - 1} x_b (\beta_j - 1)(v(ux)^{\alpha})_j^{\beta_j - 2} v_{js} \alpha_s (ux)_s^{\alpha_s - 1} x_t$$

$$+ \delta_{as} \sum_j w_{r,j} \beta_j (v(ux)^{\alpha})_j^{\beta_j - 1} v_{ja} \alpha_a x_b (\alpha_a - 1)(ux)_a^{\alpha_a - 2} x_t,$$

thus

$$\sum_{s,t} u_{st} \frac{\partial^2 f_r(x)}{\partial u_{st} \partial u_{ab}} = \sum_j w_{r,j} \beta_j v_{ja} \alpha_a (ux)_a^{\alpha_a - 1} x_b (\beta_j - 1)(v(ux)^{\alpha})_j^{\beta_j - 2} \sum_s v_{js} \alpha_s (ux)_s^{\alpha_s - 1} \sum_t u_{st} x_t$$

$$+ \sum_j w_{r,j} \beta_j (v(ux)^{\alpha})_j^{\beta_j - 1} v_{ja} \alpha_a x_b (\alpha_a - 1)(ux)_a^{\alpha_a - 2} \sum_t u_{at} x_t$$

$$= \sum_j w_{r,j} \beta_j v_{ja} \alpha_a (ux)_a^{\alpha_a - 1} x_b (\beta_j - 1)(v(ux)^{\alpha})_j^{\beta_j - 2} \sum_s v_{js} \alpha_s (ux)_s^{\alpha_s}$$

$$+ \sum_j w_{r,j} \beta_j (v(ux)^{\alpha})_j^{\beta_j - 1} v_{ja} \alpha_a x_b (\alpha_a - 1)(ux)_a^{\alpha_a - 1}$$

$$\leq \|\alpha\|_{\infty} \sum_j w_{r,j} \beta_j v_{ja} \alpha_a (ux)_a^{\alpha_a - 1} x_b (\beta_j - 1)(v(ux)^{\alpha})_j^{\beta_j - 2} \sum_s v_{js}(ux)_s^{\alpha_s}$$

$$+ \|\alpha - \mathbf{1}\|_{\infty} \sum_j w_{r,j} \beta_j (v(ux)^{\alpha})_j^{\beta_j - 1} v_{ja} \alpha_a x_b (ux)_a^{\alpha_a - 1}$$

$$\leq (\|\alpha\|_{\infty} \|\beta - \mathbf{1}\|_{\infty} + \|\alpha - \mathbf{1}\|_{\infty}) \frac{\partial f_r(x)}{\partial u_{ab}}$$

$$\Longrightarrow \qquad M_{u,u} = \|\alpha\|_{\infty} \|\beta - \mathbf{1}\|_{\infty} + \|\alpha - \mathbf{1}\|_{\infty}.$$

As $\alpha_i, \beta_j \geq 1$ for every $i \in [n_1], j \in [n_2]$ by assumption, we have $\|\alpha - \mathbf{1}\|_\infty = \|\alpha\|_\infty - 1$ and $\|\beta - \mathbf{1}\|_\infty = \|\beta\|_\infty - 1$ so that $M_{u,u} = \|\alpha\|_\infty \|\beta\|_\infty - 1$ and $M_{v,v} = \|\beta\|_\infty - 1$. Now, we look at the mixed second derivatives. We have

$$\frac{\partial^2 f_r(x)}{\partial v_{st} \partial w_{a,b}} = \delta_{ra} \delta_{bs} \beta_s (v(ux)^\alpha)_s^{\beta_s - 1} (ux)_t^{\alpha_t}.$$

It follows that

$$\sum_b w_{a,b} \frac{\partial^2 f_r(x)}{\partial v_{st} \partial w_{a,b}} = \delta_{ra} w_{a,s} \beta_s (v(ux)^\alpha)_s^{\beta_s - 1} (ux)_t^{\alpha_t} \leq w_{r,s} \beta_s (v(ux)^\alpha)_s^{\beta_s - 1} (ux)_t^{\alpha_t} = \frac{\partial f_r(x)}{\partial v_{st}}$$

$$\implies \qquad M_{v,w} = 1,$$

and

$$\sum_{s,t} v_{st} \frac{\partial^2 f_r(x)}{\partial v_{st} \partial w_{a,b}} = \delta_{ra} \beta_b (v(ux)^\alpha)_b^{\beta_b - 1} \sum_t v_{bt} (ux)_t^{\alpha_t} = \beta_b \frac{\partial f_r(x)}{\partial w_{a,b}} \leq \|\beta\|_\infty \frac{\partial f_r(x)}{\partial w_{a,b}}$$

$$\implies \qquad M_{w,v} = \|\beta\|_\infty.$$

Furthermore, it holds

$$\frac{\partial^2 f_r(x)}{\partial u_{st} \partial w_{ab}} = \delta_{ra} \beta_b (v(ux)^\alpha)_b^{\beta_b - 1} v_{bs} \alpha_s (ux)_s^{\alpha_s - 1} x_t,$$

so that

$$\sum_b w_{a,b} \frac{\partial^2 f_r(x)}{\partial w_{ab} \partial u_{st}} = \delta_{ra} \sum_b w_{a,b} \beta_b (v(ux)^\alpha)_b^{\beta_b - 1} v_{bs} \alpha_s (ux)_s^{\alpha_s - 1} x_t$$

$$\leq \sum_b w_{r,b} \beta_b (v(ux)^\alpha)_b^{\beta_b - 1} v_{bs} \alpha_s (ux)_s^{\alpha_s - 1} x_t = \frac{\partial f_r(x)}{\partial u_{st}}$$

$$\implies \qquad M_{u,w} = 1,$$

and

$$\sum_{s,t} u_{st} \frac{\partial^2 f_r(x)}{\partial w_{ab} \partial u_{st}} = \delta_{ra} \beta_b (v(ux)^\alpha)_b^{\beta_b - 1} \sum_s v_{bs} \alpha_s (ux)_s^{\alpha_s - 1} \sum_t u_{st} x_t$$

$$= \delta_{ra} \beta_b (v(ux)^\alpha)_b^{\beta_b - 1} \sum_s v_{bs} \alpha_s (ux)_s^{\alpha_s}$$

$$\leq \|\alpha\|_\infty \delta_{ra} \beta_b (v(ux)^\alpha)_b^{\beta_b} \leq \|\alpha\|_\infty \|\beta\|_\infty \frac{\partial f_r(x)}{\partial w_{a,b}}$$

$$\implies \qquad M_{w,u} = \|\alpha\|_\infty \|\beta\|_\infty.$$

Finally, we have

$$\frac{\partial^2 f_r(x)}{\partial u_{st} \partial v_{ab}} = w_{r,a} \beta_a (ux)_b^{\alpha_b} (\beta_a - 1) (v(ux)^\alpha)_a^{\beta_a - 2} v_{as} \alpha_s (ux)_s^{\alpha_s - 1} x_t$$

$$+ \delta_{sb} w_{r,a} \beta_a (v(ux)^\alpha)_a^{\beta_a - 1} \alpha_b (ux)_b^{\alpha_b - 1} x_t.$$

Hence,

$$\sum_{a,b} v_{ab} \frac{\partial^2 f_r(x)}{\partial u_{st} \partial v_{ab}} = \sum_a w_{r,a} \beta_a (\beta_a - 1) (v(ux)^\alpha)_a^{\beta_a - 2} v_{as} \alpha_s (ux)_s^{\alpha_s - 1} x_t \sum_b v_{ab} (ux)_b^{\alpha_b}$$

$$+ \sum_a w_{r,a} \beta_a (v(ux)^\alpha)_a^{\beta_a - 1} v_{as} \alpha_s (ux)_s^{\alpha_s - 1} x_t$$

$$= \sum_a w_{r,a} \beta_a (\beta_a - 1) (v(ux)^\alpha)_a^{\beta_a - 1} v_{as} \alpha_s (ux)_s^{\alpha_s - 1} x_t + \frac{\partial f_r(x)}{\partial u_{st}}$$

$$\leq (\|\beta - \mathbf{1}\|_\infty + 1) \frac{\partial f_r(x)}{\partial u_{st}}$$

$$\implies M_{u,v} = \|\beta - \mathbf{1}\|_\infty + 1 = \|\beta\|_\infty,$$

and

$$\sum_{s,t} u_{st} \frac{\partial^2 f_r(x)}{\partial u_{st} \partial v_{ab}} = w_{r,a} \beta_a (ux)_b^{\alpha_b} (\beta_a - 1)(v(ux)^\alpha)_a^{\beta_a - 2} \sum_s v_{as} \alpha_s (ux)_s^{\alpha_s - 1} \sum_t u_{st} x_t$$

$$+ w_{r,a} \beta_a (v(ux)^\alpha)_a^{\beta_a - 1} \alpha_b (ux)_b^{\alpha_b - 1} \sum_t u_{bt} x_t$$

$$= w_{r,a} \beta_a (ux)_b^{\alpha_b} (\beta_a - 1)(v(ux)^\alpha)_a^{\beta_a - 2} \sum_s v_{as} \alpha_s (ux)_s^{\alpha_s}$$

$$+ w_{r,a} \beta_a (v(ux)^\alpha)_a^{\beta_a - 1} \alpha_b (ux)_b^{\alpha_b}$$

$$\leq \|\alpha\|_\infty w_{r,a} \beta_a (ux)_b^{\alpha_b} (\beta_a - 1)(v(ux)^\alpha)_a^{\beta_a - 2} \sum_s v_{as} (ux)_s^{\alpha_s}$$

$$+ \|\alpha\|_\infty w_{r,a} \beta_a (v(ux)^\alpha)_a^{\beta_a - 1} (ux)_b^{\alpha_b}$$

$$= \|\alpha\|_\infty w_{r,a} \beta_a (ux)_b^{\alpha_b} (\beta_a - 1)(v(ux)^\alpha)_a^{\beta_a - 1} + \|\alpha\|_\infty \frac{\partial f_r(x)}{\partial v_{ab}}$$

$$\leq \|\alpha\|_\infty (\|\beta - \mathbf{1}\|_\infty + 1) \frac{\partial f_r(x)}{\partial v_{ab}}$$

$$\implies M_{v,u} = \|\alpha\|_\infty (\|\beta - \mathbf{1}\|_\infty + 1) = \|\alpha\|_\infty \|\beta\|_\infty.$$

$\square$