[Reviews · NeurIPS 2016]

Reviewer 1

Summary

This paper investigates a special class of neural networks, where the author succeed to show that it has only one stationary point. The author introduce a fast method converging to the unique fixed-point that is the global minimum of the cost function.

Qualitative Assessment

The paper investigates an interesting problem of showing a relatively complex model with wisely chosen constraints can have only one stationary point. Apart from convexity (or its generalizations like geodesic convexity), the common practice for showing uniqueness of the stationary point is through using fixed-point theory; the approach that is used in this paper. The author construct a contractive map, that its fixed-point is the unique minimizer of the cost function. One drawback of the proposed method is dependence of the convergence proof on the boundedness of spectral radius of some non-trivial matrix. As authors mention, that is the reason why they did not try networks with larger number of hidden units. Another problem with the method is many parameters that should be tuned that makes the use of the procedure rather unintuitive. Furthermore, for each choice of the parameters the spectral condition should be verified. In the experimental results, the authors tried about 150,000 different combination of possible parameters values and chose the best of them based on cross-validation accuracy. Their method, however, could outperform linear-SVM for only 2 out of 7 datasets. This result questions, the real power of the neural network they are investigating.

Confidence in this Review

2-Confident (read it all; understood it all reasonably well)


Reviewer 2

Summary

This paper studied a particular class of feedforward neural networks that can be trained globally optimal with a linear convergence rate using nonlinear spectral method. This method was applied to deep networks with one- and two-hidden layers. Experiments were conducted on a series of real world datasets.

Qualitative Assessment

This paper studied a particular class of feedforward neural networks that can be trained globally optimal with a linear convergence rate using nonlinear spectral method. This method was applied to deep networks with one- and two-hidden layers. Experiments were conducted on a series of real world datasets. As stated by authors, the class of feedforward neural networks studied is restrictive and counterintuitive by imposing the non-negativity on the weights of network and maximizing the regularization of these weights. Moreover, the less popular activation function called generalized polynomial is required for the optimality condition. All these assumptions are not quite reasonable. Detailed and careful explanations from perspectives of applications or real world datasets are helpful instead of making more restricted assumptions to achieve global optimum. Authors claimed that the targeted class of neural networks still have enough expressive power to model complex decision boundaries and achieves good performance. However, the empirical experimental results do not give the strong support for the above claim. It seems that linear SVM performs much better than NLSM1 and NLSM2 in the cases that linear SVM seems not to be properly tuned, while NLSM tune their network structures. Moreover, as we know, NLSM is a nonlinear model, it is strange why a linear SVM is used as baseline, not a nonlinear SVM such as polynomial kernel or Gaussian kernel. Thus, the comparisons are not fair, and the conclusions based on these comparable results are not convincing. In other words, these results cannot prove the enough expressive power of the studied class of neural networks. In the experiments, the datasets from UCI repo is too small from both sample size and feature dimensionality. Based on the analysis, the bounds on the spectral radius of the matrix A grow with the number of hidden units, so it is problematic for high-dimensional data. Is there any way to deal with this issue? This will greatly prevent this method from being applied to a large proportion of applications. It seems that the optimal condition is not guaranteed by any settings of parameters possibly used in the studied neural networks. Although the condition can be checked without running the neural networks, it is still problematic if we do not have any priors or the possible parameters are extremely huge. Any fast approach to get the possible small set of valid candidates? Does the optimal property of the studied model come from the special objective function? It is interesting to see if stochastic gradient descent can also converge to global optimal empirically on some tested datasets.

Confidence in this Review

2-Confident (read it all; understood it all reasonably well)


Reviewer 3

Summary

This paper presents a nice theory that demonstrates that a certain class of neural network has a global optimal which can be achieved in a linear convergence rate. The main constrain is that all the input data and the weights of the linear layer are positive and activation functions are generalized polynomial. The proposed theory is first proved on the neural network with a single hidden layer and later extend to the networks with multiple hidden layers. The authors also proposed an optimization algorithm based on nonlinear spectral method. Experiments show that the proposed algorithm works quite well on a small 2D synthetic dataset, and it also achieves a reasonable performance on some small real-world datasets (UCI datasets).

Qualitative Assessment

Pros: As claimed by the authors, this is the first work that proposed a potentially practical neural work with optimal guarantee and convergence guarantee. Previous work that have convergence guarantees requires a complicated method to check its preconditions, some of them even impossible to check in practice, as they depend on the data-generating measure. The theory proposed in this work has comparatively simple preconditions that only depends on the spectral radius of a nonnegative matrix consists of the parameters of the network. The authors also demonstrate that the proposed theory can be applied to network of arbitrary depth and show a reasonable result on small real dataset. Cons: The proposed network still has limited applications. The theory requires both the input data and network parameters be positive, activation functions be generalized polynomials, and spectral radius (Theorem 1) be smaller than 1. I think the non-negative constrains on the input data should not block too much potential usage, as most of train data can be easily shift to be all positive. However, the fact that network parameters are positive might limit the expressiveness of the model. Also, the fact that all activation functions of all hidden layers are different is unnatural and it might also limit its application. At last, the authors said that in order to satisfy the spectral radius constrains, they need to increase p1 and p2 and decrease pw and pu, which effectively decreases the weights of linear units. And if I understand correctly, in order to satisfy the spectral radius constrains, the higher dimension of the input and network parameters, the lower the weights of linear units. Since when the linear weight is very small, the activation functions will almost act the same as a linear function. In that case, adding more layers is just adding another linear function, which won't increase the expressiveness of the network. Thus, I am little worried that this method might not easy to extend it to deep architectures. Despite all these concerns, this is still an encouraging result as an initial attempt and I think this paper should be accepted. Minor comments: I am curious the how p1, p2, pw, pu, the dimension of the input and the size of the network parameters affects the spectral radius rho(A). For example, when the dimension of input increases, does rho(A) increases linearly or exponentially, and in order to counter effect that, should we linearly or exponentially decreases pw and pu. I know it might not be easy to prove that, but it might be possible to empirically plot that relationship on a synthetic data.

Confidence in this Review

1-Less confident (might not have understood significant parts)


Reviewer 4

Summary

The paper presents nonlinear spectral method which optimally trains a particular class of feedforward neural networks with a linear convergence rate. The condition which guarantees global optimality depends on the parameters of the architecture of the network and boils down to the computation of the spectral radius of a small nonnegative matrix.

Qualitative Assessment

It seems new and interesting to propose a nonlinear spectral method which optimally trains a particular class of feedforward neural networks. At the same time, I'm worried about the limitation of the approach; it imposes the condition of non-negativity on the weights of the network. In addition, the choice of the activation functions in the neural network are non-standard and very specific ones. The limitation seems very strong, which may hinder the good performance of neural networks. Indeed, numerical experiments show not-so-good performance over SVM. Minor comment: L.90-91: The sentence saying "Note that the nonlinear spectral method has a linear convergence rate and thus converges quickly typically in a few (less than 10) iterations to..." seems exaggerated because the linear convergence rate does not necessarily imply a few iterations.

Confidence in this Review

1-Less confident (might not have understood significant parts)


Reviewer 5

Summary

The paper discusses and defines a specific kind of feedforward neural networks for which a unique global optimizer can be obtained. As a product, it defines the algorithm that finds the global optimizer and the algorithm has a linear convergence rate.

Qualitative Assessment

The mathematical derivation is instructive although a few parts are not very clear to me. 1. Theorem 3 involves the computation of first order derivatives as shown in those inequalities, but Theorem 4 gets to the second derivative level. What causes that? 2. The design of this kind of network seems to treat the last layer separately from any hidden layer. Since u is also a matrix representing the parameters connecting to the hidden layer, the gradient with respect to u in G^\Phi (3) is just a gradient in terms of the vectorization of u? (Similar questions arise in several other places, like bounding u to be within a ball). A concern (do I miss anything?) is that why the model needs to maximize the 1-norm of the connection parameters in (2). In most NN models, weight decay requires to reduce the 1-norm. It appears that this is because the arbitrarily small epsilon in the (2) is just needed to make the gradient strictly positive so it's in V_++. This sounds just needed for their proof but does not justify well in practical NNs. What warrants that there is no free parameter here? It requires all the data to be non-negative and then the model uses all positive weights (not even zero), so it keeps adding up. How practical is this design? In the experiments, the proposed NN is compared with SVM (because SVM also has the global optimizer?). However, in practice, will this kind of NNs brings additional values in some aspect than other NNs (those more commonly used NNs, e.g., with ReLU)?

Confidence in this Review

2-Confident (read it all; understood it all reasonably well)


Reviewer 6

Summary

The paper proposes a method to train -under certain assumptions- a specific type of feed-forward neural networks. The authors prove optimal convergence and demonstrate that their method can be applied on a selection of "real world" problems.

Qualitative Assessment

The problem under investigation seems interesting to me and any theoretical advance in this direction is appreciated. But the paper falls a bit short in convincing that the proposed class of neural networks is useful in practice. In particular, if the authors raise the issue of the expressiveness of their restricted model (and they do so twice), then I would recommend adding some thoughts about what can be done and what is out of reach for their approach. Moreover, concurrent approaches should be discussed and contrasted. The experimental results do not clarify the limitations of applicability of their model either. When reading the experimental results, I was surprised that the authors only compare their approach to some SVM and ignore other methods, both other training methods for some neural networks and other algorithmic paradigms. Also, I would have expected some information about the computational resources demanded by the three algorithms. The introduction is well written, subsequently the presentation has some room for improvement. In particular, Section 2 does not read fluently. There are some typos, I have collected a few below. Some specific remarks: While the definition of the loss function is straight-forward, the function $\Phi$ lacks motivation: the role of the epsilon term was explained in the rebuttal, I think this should be added to the paper as well. Theorem 1 should be explained informally before being stated. The statement itself is hard to parse. Below Thm 1: The authors explained in the rebuttal that the very general statement that "the nonlinear spectral method [...] converges quickly typically in a few (less than 10) iterations to the global maximum", given immediately after Theorem 1, stems from their experimental observation. This should be clarified. 60: polyomial 67: one CAN model. 67: p_1 and p_2 are not introduced. 67: R_{++} is not introduced, which is unfortunate since there are several other variables whose definition points to that one. Thanks for clarifying this in the rebuttal, I had not seen this notation before.

Confidence in this Review

1-Less confident (might not have understood significant parts)